# A novel channel reduction concept to enhance the classification of motor imagery tasks in brain-computer interface systems

Taslima Khanam[1]*, Siuly Siuly[1], Kabir Ahmad[2], Hua Wang[1]

1 Institute for Sustainable Industries and Liveable Cities, Victoria University, Melbourne, Australia,
2 Barwon Health, Geelong, Australia

* taslima.khanam@live.vu.edu.au

## Abstract

Electroencephalogram (EEG) signals play a critical role in advancing brain-computer interface (BCI) systems, particularly for detecting motor imagery (MI) movements. However, analysing large volume of EEG datasets faces some challenges due to redundant information, and performance degradation. Irrelevant channels introduce noise, which reduces accuracy and slows system performance. To address these issues, this study aims to develop a novel channel selection method to enhance EEG-based MI task performance in BCI applications. Our proposed hybrid approach combines statistical t-tests with a Bonferroni correction-based channel reduction technique, followed by the application of a Deep Learning Regularized Common Spatial Pattern with Neural Network (DLRCSPNN) framework. This framework employs DLRCSP for feature extraction and neural network (NN) algorithm for classification. Our developed method excluded channels with correlation coefficients below 0.5, retaining only significant, non-redundant channels and tested on three real-time EEG-based BCI datasets. This study produces the highest accuracy score in the case of every subjects above 90% for all the applied datasets. In the first dataset, our method achieved the highest accuracy, improving by 3.27% to 42.53% in terms of individual subject compared to seven existing machine learning algorithms. In the second and third dataset, it outperformed existing approaches, with accuracy gains of 5% to 45% and 1% to 17.47% respectively. Comparisons with a CSP and NN framework confirmed DLRCSPNN's algorithms superior performance. These results demonstrate the effectiveness of the approach, offering a new perspective on the identification of MI task performance in EEG based BCI technology. This proposed technique will enable rapid identification of motor-disabled individuals' intentions, supporting patient rehabilitation and improving daily living.

**Data availability statement:** This study uses publicly available EEG datasets from the BCI competitions. The authors cannot redistribute the raw data but access is freely available from the official competition website (http://www.bbci.de/competition/) or by contacting the organizers at bci-competition@bbci.de. Specifically, we used BCI Competition III, data set IVa (http://www.bbci.de/competition/iii) and BCI Competition IV, data Sets 1 and 2a (http://www.bbci.de/competition/iv). Permission to use these datasets was obtained under the terms set by the competition organizers.

**Funding:** The author(s) received no specific funding for this work.

**Competing interests:** The authors have declared that no competing interests exist.

## 1. Introduction

Motor disability is a neurological condition that impairs an individual's ability to move, maintain balance, and posture. In severe cases, patients may experience a locked-in state, rendering physical movement extremely challenging. A ground breaking advancement in assisting such patients is Brain-Computer Interface (BCI) technology which enables direct communication between the brain and external devices, such as computers or robotic systems, bypassing the need for muscular control [1–5]. These systems translate brain activity into control signals that can operate external devices or provide feedback to the brain. Once brain signals are acquired, they are processed and decoded using signal processing algorithms and machine learning techniques [6,7]. The decoded signals can then control devices like computer cursors, robotic arms, or wheelchairs, helping individuals with motor disabilities regain the ability to communicate or manage their environment. The core principle of BCI systems is to record and analyze brain activity to extract meaningful information that supports the treatment and rehabilitation of those with motor impairments [8–12]. These BCI systems often rely on motor imagery (MI), where users imagine movements without physical execution. MI tasks are identified by analyzing brain signals captured through Electroencephalography (EEG), a technique that records electrical brain activity via scalp electrodes [13–17]. However, EEG data management is complex due to its non-invasive, nonlinear, non-stationary, non-Gaussian, and noisy nature [4,18]. Additionally, EEG signals are highly subject-specific, making model generalization across individuals challenging. A key issue is the high dimensionality of multichannel EEG signals, which can negatively impact classification accuracy (CA). Channel selection methods help mitigate this by identifying the most relevant EEG channels for a given task, reducing redundant data, and maintaining CA while lowering computational complexity [19–25].

Recent advancements in BCI research focus on optimizing channel selection to enhance classification performance. Khalid et al. proposed a method that combines triple-shallow convolutional neural networks (TSCNN) with a deep genetic algorithm fitness formation (DGAFF) for channel selection, achieving subject-wise accuracy between 73.41% and 97.82% [26]. But with model complexity and data dependency issues. Vadivelan and Sethuramalingam introduced a double-branch EEGNet (DB-EEGNET) with a multi-objective prioritized jellyfish search (MPJS) algorithm, reporting 83.9% accuracy, though facing performance inconsistencies [27]. Qin et al. developed a cross-domain-based channel selection (CDCS) approach using common spatial patterns (CSP) for feature extraction and linear discriminant analysis (LDA) for MI task identification, achieving 77.57% and 66.06% accuracy on BCI competition datasets but suffering from limited trial data and public template constraints [28]. Other studies explored different approaches: Lee et al. applied a NeuroXAI-based channel selection approach, integrated with EEGNet but faced limitations like high computational costs and inconsistent results [29]. Dovedi et al. used a channel selection method based on the ReliefF algorithm and used the Local Maximum Synchro-Squeezing Transform (LMSST) for feature extraction, reporting a minimum accuracy of 79.85% on the BCI competition IV (2a) dataset, but at high computational expense [30]. A

CSP- based on rank channel selection for multifrequency band EEG (CSP-R-MF) achieved 77.75% accuracy but was limited by frequency band dependency and complex signal decomposition [31]. Frequency band dependency and complexity in signal decomposition are the limitation of this method. Additionally, the author in identified optimal channels using a combination of class correlation, random forest-based feature ranking, and the Infinite Latent Feature Selection (ILFS) algorithm. The challenge of this method is individual variability, computational demands, and frequency band specificity [32]. Other methods, including random fisher discriminant analysis (FDA) with a multiobjective hybrid real-binary Particle Swarm Optimization, forest-based feature ranking, filter-bank CSP, and genetic algorithms, also showed promise but suffered from issues like overfitting, high computational demands, and long convergence times [33–38]. Du et al. proposed a channel selection strategy based on brain function networks, introducing Synchronization Likelihood as a connection index to build a motor imagery brain functional network. The centrality analysis of this network was then used to select a combination of strongly motor-related leads. However, the complexity of this method resulted in no subject achieving more than 97% accuracy [39].

Despite these advancements, existing methods often require significant computational time, exhibit low classification accuracy, or fail to generalize across subjects. Effective channel selection should reduce computational complexity, prevent overfitting, and improve classification accuracy while minimizing EEG setup time. To address these challenges, we propose an AI-based channel selection method integrating statistical tests with Bonferroni correction-based channel reduction. This hybrid approach enhances accuracy in EEG-based MI task classification. In this study, we employ a regularized common spatial patterns technique with deep learning (DLRCSP), where the covariance matrix is shrunk toward the identity matrix, and the γ regularization parameter is automatically determined using Ledoit and Wolf's method [40].

Therefore, the goal of our study is to create AI-based channel selection algorithms that are both fast and accurate enough to identify EEG-based MI movement. Hence, the contribution of this paper can be described as follows.

1) A novel hybrid approach of combining statistical tests with Bonferroni correction analysis based channel reduction rate approach was developed to select MI task-related EEG channels.

2) The design and validation of a new efficient and automatic deep learning based framework for MI task classification.

3) Extensive experiments conducted to verify the effectiveness of the proposed method.

The structure of the paper is as follows: Section 2 details the resources and methodologies employed in the proposed work. Section 3 presents experimental evidence showcasing the hybrid model's superiority in terms of mean classification accuracy and channel reduction rate. In Section 4, we compare our developed model with existing state-of-the-art models. Finally, Section 5 concludes the study and explores potential future directions in this field.

## 2. Materials and methods

### 2.1. Proposed framework

This study proposes a DLRCSPNN technique, integrating statistical tests and a Bonferroni correction-based channel reduction approach. Initially, channels were selected using a t-test and p-value, discarding those with correlation coefficients below 0.5 to ensure statistical significance and minimize redundancy. Feature extraction was performed using DLRCSP, followed by classification using neural networks (NN) and recurrent neural networks (RNN). The framework consists of five steps: (i) EEG data acquisition, (ii) channel selection, (iii) pre-processing, (iv) feature extraction, and (v) classification. A comparative framework (CSPNN) using traditional CSP for feature extraction and NN for classification was also developed. Fig 1 illustrates the methodology, with further details in subsequent sections.

### 2.2. EEG data acquisition

In our study, we utilized three publicly available datasets, dataset 1: BCI Competition III Dataset IVa and dataset 2: BCI Competition IV- dataset 1 [41,42]. All datasets are accessible via the following links, Data Set IVa for the BCI Competition

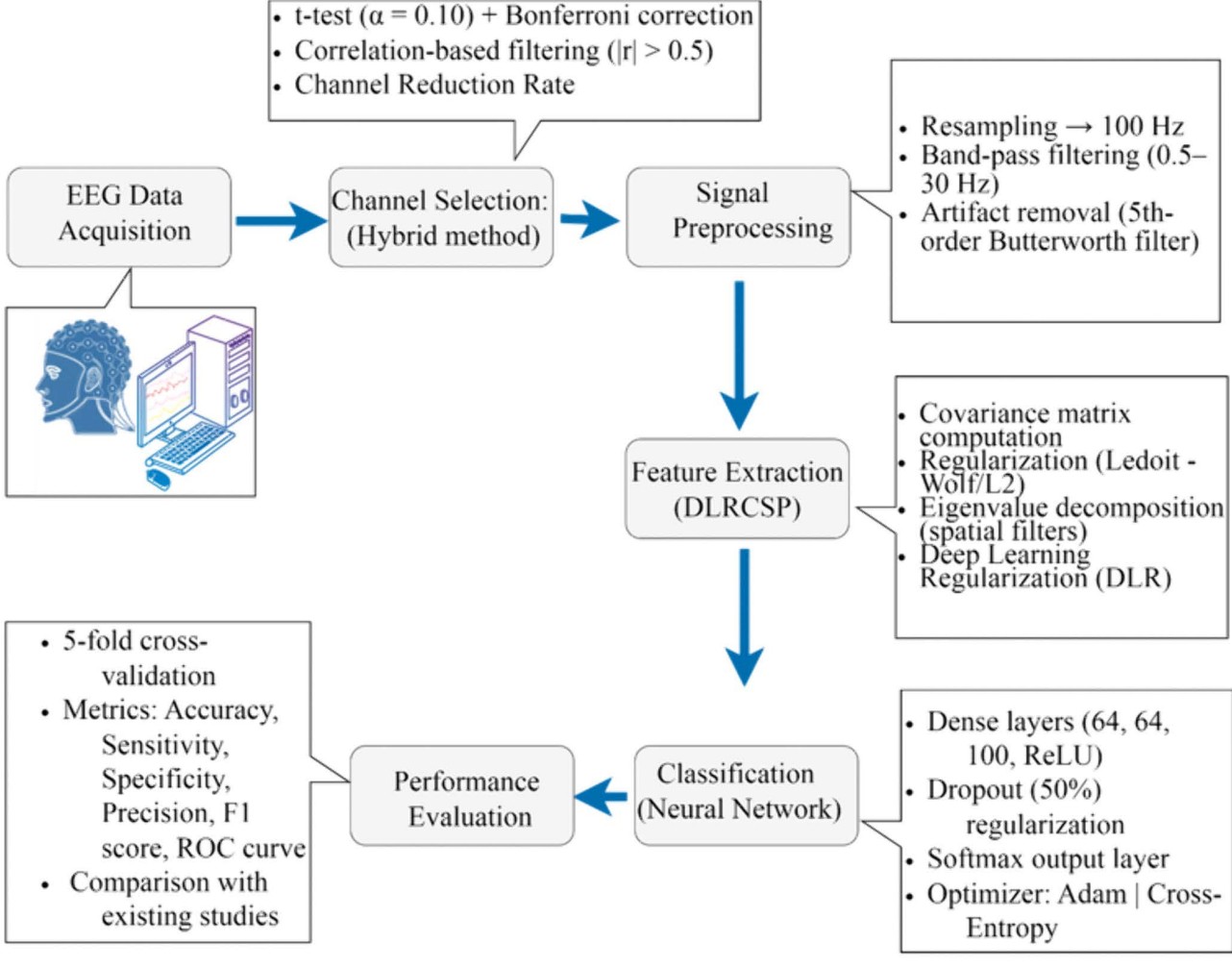

**Fig 1. An overview of the proposed framework.**

III (bbci.de), Data Set 1 for the BCI Competition IV (bbci.de) and BCI Competition IV (bbci.de). Access to these datasets is granted upon agreeing to the terms and conditions outlined in the "Download of data sets" section. We followed the same procedure to obtain access for this study. This study involves secondary data analysis of publicly available, de-identified data. In the original data collection, all participants provided informed, written consent for the use of their data in research under the assurance of confidentiality. Additionally, the datasets were fully anonymized by the original data providers before being made accessible, ensuring no identifiable information was included. As a secondary analysis, no participants in our study are directly connected to our research team, and no additional consent was required for our specific use. Therefore, this research does not require additional ethical approval. Dataset 1 was gathered from five healthy, labeled subjects (aa, al, av, aw, and ay) who performed motor imagery (MI) tasks involving their right hands (class 1) and right feet (class 2) forming a binary classification problem. The subjects were seated in a comfortable chair with armrests during the trials. The dataset includes EEG-based MI data from the first four sessions without feedback, recorded using 118 electrodes according to the 10/20 international system. Each subject participated in 140 out of a total of 280 trials, performing

one of the two MI tasks (right hand or right foot) for 3.5 seconds per trial. The training set consisted of 168, 224, 84, 56, and 28 trials for subjects aa, al, av, aw, and ay, respectively, with the remaining trials forming the test set.

Dataset 2, sourced from BCI Competition IV, also represents a binary classification setup. It includes EEG data from 59 channels across 7 participants. Two participants performed left hand (L) and feet (F) MI tasks, while the other participants engaged in right hand (R) and left hand (L) activities. During the first two runs, visual cues corresponding to left, right, or down arrow keys were presented on the screen for 4 seconds, guiding the subjects to execute the appropriate MI tasks. These visual cues were followed by a fixed cross and 2 seconds of a blank screen. The recordings were captured using an Ag/AgCl electrode cap, with a sampling rate of 100 Hz, and the calibration data consisted of two runs, each containing 100 single observations.

Dataset 3, BCI Competition IV dataset 2a, involves motor imagery tasks (left/right hand, both feet, and tongue) categorized into four classes. The dataset includes EEG data from 9 participants across two sessions, with each session consisting of 288 trials divided into six runs. One session provides labeled data for model training, while the other offers unlabeled data for evaluation. Participants were seated comfortably, with trials beginning with a fixation cross and an auditory warning. EEG signals were recorded using 22 Ag/AgCl electrodes and three EOG channels, sampled at 250 Hz, with filters applied to remove noise. Signal analysis focuses on motor imagery sequences. For analytical purposes, in this study, we extracted a binary subset by selecting only right hand (Class 1) and feet (Class 2) trials to maintain consistency with the other datasets and align with our binary classification framework.. Therefore, for all analyses presented in this study, each dataset was structured and processed for binary classification tasks. All datasets are publicly available online, and informed consent for the publication of data was obtained from each participant at the time of collection. As no personal identification information was included, no ethical approval was required for our study. Table 1 summarizes the demographic data of participants across the datasets, and Fig 2 illustrates an example pattern of EEG signals from channel C3 of both datasets after applying Butterworth filter and our developed channel selection method. The signals exhibit characteristic fluctuations in amplitude over time, with values constrained within the −50–50 range, for dataset 1 and −20–20 range for dataset 2, indicating effective filtering.

**2.2.1. Step 1: Data reduction through channel selection.** We initially applied our proposed channel selection method, a hybrid approach that combines statistical testing with correlation-based selection. This method is aims to identify the most relevant EEG channels by leveraging the significance of differences between motor imagery classes and the correlation between EEG channels. First, a t-test is conducted to evaluate the differences in EEG signals between two MI classes for each channel, using an alpha level of 0.10. The resulting p-values indicate the significance of these differences. To ensure robust results, a Bonferroni correction is applied to adjust for multiple comparisons, thereby minimizing the risk of false positives. Following this, correlation coefficients between all pairs of EEG channels are calculated to measure the degree of linear relationship between channels. Channels with high correlation values, indicating redundancy, are deprioritized in the selection process. Instead, the emphasis is placed on channels that are both statistically significant and exhibit meaningful correlations with others. The final channel selection is determined by identifying those with p-values below the threshold that also meet the strong correlation criterion (e.g., $|r| > 0.5$). We also measured channel reduction rate (CRR) for computational purposes to reduce the dimensionality of the EEG data and remove channels that may not contribute significantly to the analysis or contain artefacts or noise. In this method CRR

**Table 1. Demographic data of the used datasets.**

|  | Subjects | Channels | Trials | Frequency | Wave range |
|---|---|---|---|---|---|
| Dataset 1 | 5 | 118 | 280 | 100Hz | 0.05Hz to 200 Hz |
| Dataset 2 | 7 | 59 | 200 | 250Hz | 0.5Hz to 100 Hz |
| Dataset 3 | 9 | 22 | 288 | 250Hz | 0.5Hz to 100 Hz |

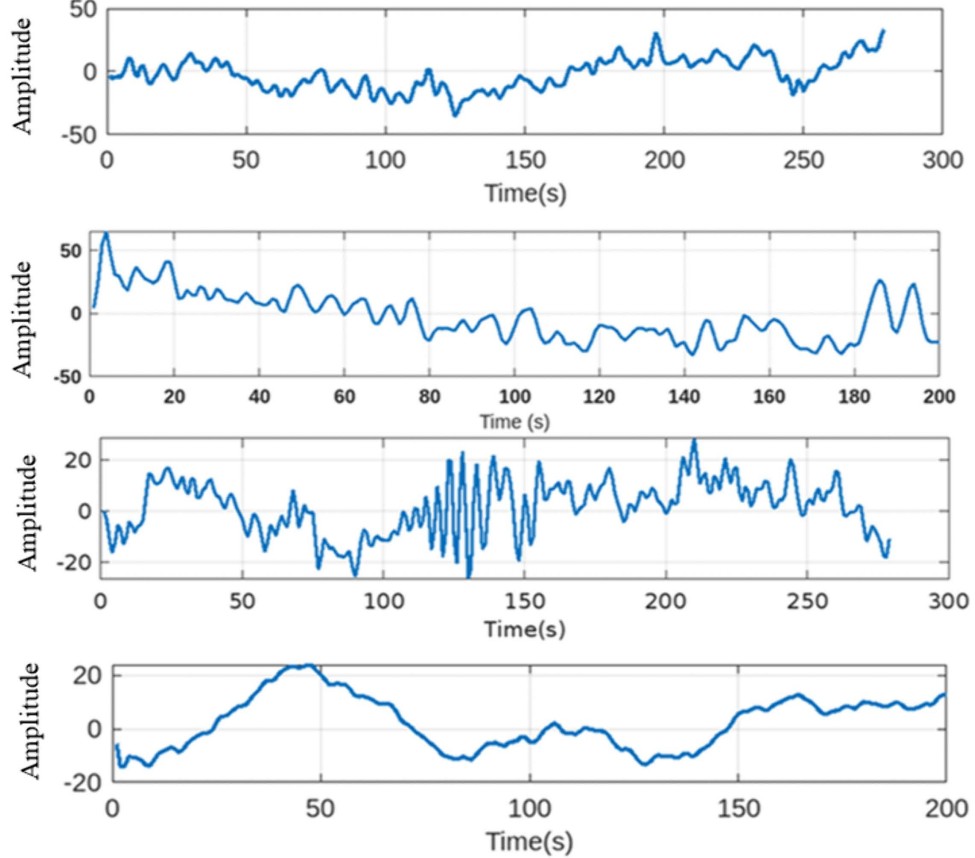

**Fig 2. An exemplary 300 sample patterns of EEG signal data from one channel (C3) of subject av, al from dataset 1 and subject a, e from dataset 2.**

was computed by using the following equation CRR = 1 – (number of selected channels/total number of channels). Thus a novel hybrid approach of combining statistical tests with Bonferroni correction analysis based channel reduction rate approach ensures that the selected channels provide statistically significant, diverse, and non-redundant information. The novelty of this method lies in its integration of statistical significance and correlation analysis, offering a more informed and efficient way to select EEG channels.

**2.2.2. Step 2: Signal pre-processing.** Following channel selection, the raw EEG data was pre-processed by standardizing the datasets to ensure comparability. Since the datasets used were originally recorded at different sampling frequencies, it was necessary to convert the EEG signals to a common sampling frequency [25]. We resampled all the datasets to 100 Hz to achieve this standardization. Additionally, we applied a frequency range of 0.5 Hz to 30 Hz to the datasets. This frequency range includes several important wave bands that are critical for analyzing brain activity: the Delta band (0.5 to 4 Hz), associated with deep sleep and unconscious processes; the Theta band (4–8 Hz), linked to drowsiness, meditation, and memory functions; the Alpha band (8–12 Hz), related to a relaxed, wakeful state, particularly with closed eyes or during meditation; and the Beta band (12–30 Hz), which is associated with active thinking, concentration, and motor activities. These bands together offer a comprehensive understanding of various mental states and cognitive functions within this frequency range [5,43].

EEG data is inherently non-linear, non-stationary, and complex, often containing artifacts that can interfere with accurate diagnosis. Some artifacts resemble signal patterns associated with neurological disorders, leading to biased interpretations during clinical assessments. Therefore, after standardizing the datasets into a common format, the next crucial step is to remove these artifacts from the raw EEG data. In this study, we utilized a Butterworth filter with 5th order derivatives to achieve this [16]. The Butterworth filter, first developed by British engineer Stephen Butterworth in 1930, offers two main advantages: a flat frequency response in the pass band and a steeper attenuation slope near the cut-off frequency as the filter order increases. By applying the Butterworth filter with a low frequency of 0.5 Hz, a high frequency of 30 Hz, and 5th order derivatives, we successfully eliminated noise from the EEG signals, resulting in clean, artifact-free data [5].

**2.2.3. Step 3: Feature extraction by DLRCSP algorithm.** We applied DLRCSP feature extraction algorithms for collecting important features from the selected channels. The Deep Learning Regularized Common Spatial Pattern (DLRCSP) algorithm is an enhancement of the traditional Common Spatial Pattern (CSP) method, incorporating deep learning techniques to improve the performance of spatial filters, particularly when dealing with noisy or high-dimensional EEG data [44,45]. The DLRCSP process begins with covariance matrix computation, where for each class c (e.g., class 1 and class 2), the covariance matrix $C_C$ of the EEG signal is computed from the data matrix $X_c$ where n represents the number of channels and m represents the number of time samples. Next, the covariance matrix is regularized using a shrinkage method, typically based on Ledoit-Wolf shrinkage or L2 regularization. This helps in stabilizing the covariance matrix by shrinking it towards the identity matrix, with a regularization parameter α determined through cross-validation. The regularized covariance matrix is then used for eigenvalue decomposition to extract the spatial filters. The traditional CSP method maximizes the variance of one class while minimizing the variance of the other class, and the spatial filters are computed by considering the covariance matrices of both classes. The DLRCSP further enhances this by integrating deep learning into the process. A deep neural network (DNN) is used to learn optimal representations of the spatial patterns from the filtered EEG data, effectively improving the regularization of the spatial filters and reducing noise in the feature extraction process. The DNN's learning process minimizes a loss function that combines the reconstruction error of the EEG data using the spatial filters, with an additional L2 regularization term to prevent overfitting and ensure generalization.

Here's a mathematical outline of DLRCSP:

**Covariance Matrix Computation**: For each class c ∈ {1,2}, the covariance matrix $C_C$ of the EEG signal is computed from the data matrix $X_C \in \mathbb{R}^{n \times m}$, where n is the number of channels and mmm is the number of time samples for class c:

$$\boldsymbol{C_C}\ \frac{1}{m}X_c X_c^T$$

(1)

Where $X_c$ is the EEG signal for class c.

**Regularization of covariance matrix**: The covariance matrix is then regularized using a shrinkage method, often based on Ledoit-Wolf shrinkage or L2 regularization, which shrinks the covariance matrix $C_C$ towards the identity matrix III. The regularized covariance matrix $C_c^{reg}$ *is computed as*:

$$C_c^{reg} = \alpha\boldsymbol{C_C} + (1-\alpha)\boldsymbol{I}$$

(2)

where α is a hyperparameter controlling the degree of shrinkage. Typically, α is determined through cross-validation.

**Eigenvalue decomposition**: Next, the **eigenvalue decomposition** of the regularized covariance matrix $C_c^{reg}$ is computed to obtain the spatial filters. Let $W_c^{reg}$ be the matrix of eigenvectors (spatial filters) corresponding to the eigenvalues $\lambda_C$.

$$C_c^{reg} W_c^{reg} = W_c^{reg}\Lambda_C$$

(3)

Where $\Lambda_C$ is the diagonal matrix of eigenvalues.

***Common spatial pattern (CSP):*** The **CSP** method involves finding spatial filters that maximize the variance of one class while minimizing the variance of the other. The spatial filters $W_c^{CSP}$ are computed by considering both classes' covariance matrices:

$$C_1^{reg} W_1^{reg} \; = \; W_1^{reg} \Lambda_1 \tag{4}$$

$$C_2^{reg} W_2^{reg} \; = \; W_2^{reg} \Lambda_2 \tag{5}$$

***Deep learning integration:*** In DLRCSP, deep learning is used to improve the regularization of the spatial filters and reduce noise in the feature extraction process. A deep neural network (DNN) is employed to further process the spatial filters obtained from the CSP step. The DNN learns optimal representations of the spatial patterns from the filtered EEG data. The learning process aims to minimize the following loss function:

$$\mathrm{L}(W_{DLRCSP} , \; X, \; y) = \; \|X W_{DLRCSP} - y\|^{\;2} + \lambda \, \|W_{DLRCSP}\|_2^2 \tag{6}$$

Where $X$ is the EEG data matrix, y ris the target labels for classification, $W_{DLRCSP}$ represents the deep learning-regularized spatial filters, $\lambda$ is the regularization parameter controlling the magnitude of the L2 norm to prevent overfitting [40].

**Step 4: Classification**: Once the features are extracted, they are classified using a Neural Network (NN) model. The NN classifier, known for its abilityin handling image-related tasks, is particularly effective at learning and classifying features while being less sensitive to noise [46]. Normally, NN is less sensitive to noise and can capture useful information from noisy data [47]. We have developed a NN model to perform classification on the generated dataset. Details of the proposed model are given below.

The proposed NN model in this study consists of multiple dense layers with dropout for regularization. Specifically, it includes three dense layers: the first two with 64 neurons each, followed by dropout layers with a 50% rate to prevent overfitting. The third dense layer has 100 neurons, and all dense layers use the ReLU activation function to introduce non-linearity. The final classification layer has 2 output units corresponding to the binary classification task and uses the softmax activation function to produce probability scores. The model is trained using the categorical cross-entropy loss function and optimized with the Adam optimizer, known for its adaptive learning rate. This architecture is designed to balance model complexity with generalization, ensuring effective performance in classifying the extracted features (Fig 3). Structural overview of the proposed NN model, where it consists of two dense layers with 64 and 32 neurons, using ReLU activation. Dropout (0.5) regularization is applied, with a softmax output layer for binary classification.

**Step 5 Performance evaluation:**

To assess the model's performance and minimize the risk of overfitting, cross-validation is employed, as high classification rates might depend on specific training and testing sets. K-fold cross-validation is a common method for this purpose, where the dataset is randomly split into k subsets of equal or nearly equal size [48]. The model is trained on k-1 subsets, and the remaining subset is used for testing. This process is repeated k times, ensuring each subset is tested once. In our study, we utilized 5-fold cross-validation to evaluate the model's performance. Additionally, we compared our proposed model against another popular neural network model, the Recurrent Neural Network (RNN), to determine their relative performance in binary-class classification.

**Recurrent Neural Network RNN**: It is a type of neural network that were designed to work with sequential data, such as time series or natural language.

RNNs were introduced by David Rumelhart and Geoffrey Hinton in 1986. For the RNN model, we used an architecture with 16 LSTM units in the first layer, followed by a dense layer with 32 neurons, and additional dropout layers to prevent

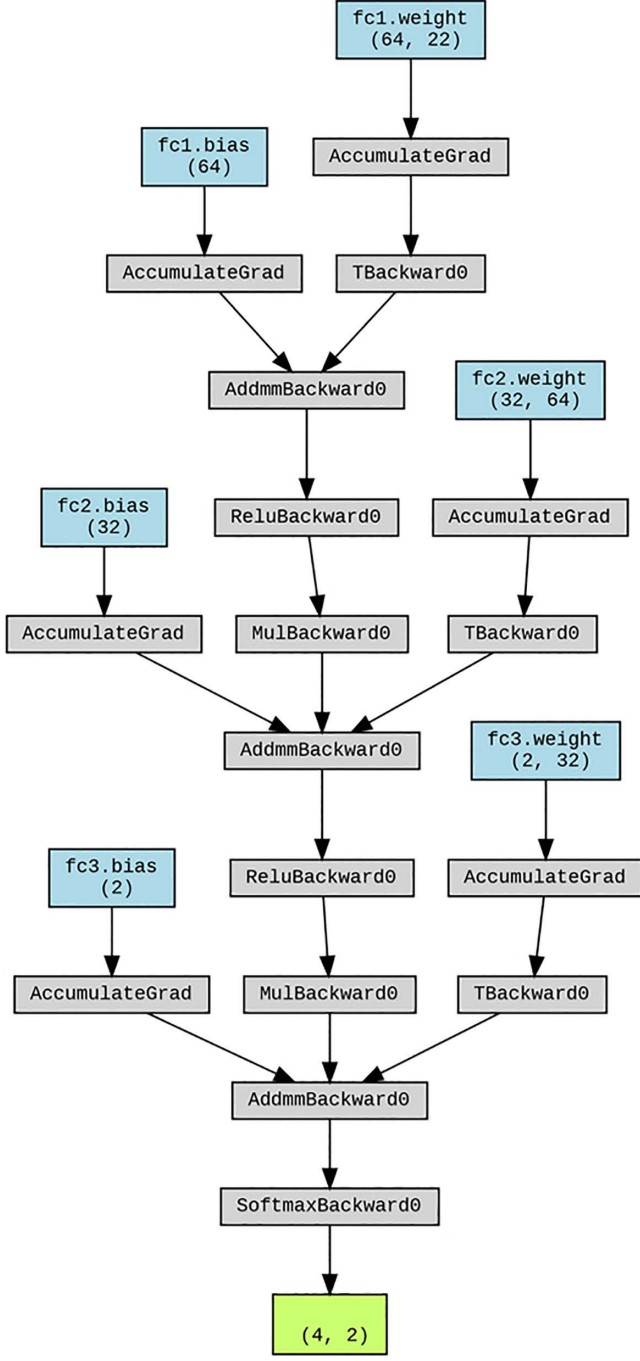

**Fig 3. Structural overview of the proposed NN model.**

overfitting. The LSTM layers are followed by a dense layer with 32 neurons, using ReLU activation, and another dropout layer. The final output layer contains 2 neurons, with softmax activation for binary classification. The model is trained using the categorical cross-entropy loss function and the Adam optimizer. This architecture ensures that the number of neurons and the sequence of layers are consistent between the NN model and RNN model, allowing for a fair and meaningful

comparison. LSTM-based RNNs are chosen due to their ability to capture long-range dependencies in sequential data, making them well-suited for EEG signal processing tasks. While GRUs could offer faster computation, LSTMs were selected for their proven effectiveness in EEG classification tasks. The decision to focus on LSTM models ensures a consistent evaluation framework for this study. Future research could explore comparisons with GRUs or other architectures to further validate the approach [49]. We evaluated the models using seven parameters: (1) Channel Reduction Rate (CRR), (2) Classification Accuracy (CA), (3) Precision, (4) Sensitivity, (5) Specificity, (6) F1-Score, and (7) Receiver Operating Characteristic (ROC) curve.

CRR in motor imagery EEG data refers to the percentage or proportion of channels or electrodes that are discarded or removed during pre-processing or analysis of the EEG data. Motor imagery EEG data typically includes multiple channels or electrodes placed on the scalp to capture electrical brain activity associated with MI tasks. Sensitivity measures the model's ability to correctly identify positive class instances, while specificity measures its ability to correctly identify negative class instances. Classification Accuracy (CA) evaluates the correctness of the model in classifying different MI activities based on EEG data, representing the percentage of correctly classified instances. Precision evaluates the accuracy of positive predictions, while recall assesses the completeness of positive predictions. F1 score combines precision and recall to measure the overall accuracy of the model. It combines the precision and recall scores of a model. The accuracy metric computes how many times a model made a correct prediction across the entire dataset. Finally, ROC graph is a highly valuable tool for visualizing the reliability of a classifier, created by plotting sensitivity (true positive rate) on the Y-axis against 1-specificity (false positive rate) on the X-axis [4,50].

***Experimental setup:*** In the preprocessing stage, EEG data from each subject was segmented into 3-second intervals. Channel selection was performed using t-tests, with p-values corrected via Bonferroni adjustment. Channels with significant p-values and correlation coefficients above 0.5 were retained. For dataset 1, subjects aa, al, av, aw, ay had reduced channels (RC) of 49, 93, 47, 27, and 27 with corresponding CRR of 0.41, 0.78, 0.39, 0.22, and 0.22. For dataset 2, subject a, b, c, d, e, f, g had RC values of 37, 31, 31, 22, 26, and 40 with CRRs of 22 0.61, 0.52, 0.52, 0.37, 0.44, 0.67, 0.37. For dataset 3, subjects A01, A02, A03, A04, A05, A06, A07, A08, A09 had RC values 8, 6, 11, 12, 6, 5, 11, 11, and 8 with CRRs 0.36, 0.27, 0.5, 0.54, 0.27, 0.22, 0.5, 0.5, 0.36. After pre-processing stage, it generated 301x (RC) x280 samples for each subject of dataset 1, 301x (RC) x200 for each subject of dataset 2 and 301x(RC)x288 for each subject of dataset 3. After applying DLRCSP, the number of sample for each subject of dataset 1 was 200x (RC), for each subject of dataset 2 was 280x (RC) and for dataset 3 was 288x(RC) with binary classes. After sample generation, the generated samples were randomly divided into five equal or nearly equal parts for 5-fold cross-validation of different DLRCSP models. Each experimental model was trained on four parts while the fifth part was used for validation, ensuring that each part was used for validation exactly once. To address overfitting in the Neural Network (NN) and Recurrent Neural Network (RNN) classifiers, several techniques were implemented to enhance generalization and prevent the model from memorizing training data. K-Fold Cross-Validation (5-fold) was applied to ensure the model is trained and validated on different data splits, reducing variance and improving robustness. Feature selection using statistical tests (t-tests) and correlation thresholding helped remove less significant and highly correlated EEG channels, minimizing redundant and noisy features that could lead to overfitting. Dropout layers (0.5 rate) were introduced in the neural network to randomly deactivate neurons during training, preventing reliance on specific features and promoting generalization. Additionally, data standardization using StandardScaler ensured uniform feature scaling, preventing dominance by high-magnitude features and enabling stable model convergence. The Adam optimizer with an implicit L2 penalty was used for weight regularization, preventing large weight updates that could overfit the data. Furthermore, performance monitoring through standard deviation calculations across folds, ROC curves, and AUC analysis was included to detect overfitting and ensure the model maintains balanced performance on both training and test data. To compare the accuracies of DLRCSPNN and DLRCSPRNN, a paired t-test was conducted, as it is specifically designed for comparing two related datasets—where the same subjects are evaluated under different conditions. Unlike ANOVA, which is used for multiple models, or an independent t-test, which is suited

for unrelated groups, the paired t-test provided the most appropriate statistical comparison in this study. All models were trained for 100 epochs using mini-batch mode for efficient learning. Batch sizes of 32, 64, and 128 were tested to optimize performance. The experimental analysis was conducted on Google Colab, utilizing 2 CPU cores, 12.67 GB of RAM, and no GPU acceleration.

## 3. Result

This section presents the experimental results of the proposed system. In this study, we compared the developed model with the DLRCSP-based RNN model, using 5-fold cross-validation. Both models were trained for 100 epochs to avoid overfitting and we examined the impact of batch sizes [32, 64, and 128] on performance using sensitivity, specificity, precision, F1 score, and accuracy. Tables 2 and 3 summarize the classification accuracy, selected channels, and channel reduction rate (CRR) for the DLRCSPNN and DLRCSPRNN models.

Accuracy varied across subjects with different batch sizes, over the 5-fold cross validation. For batch size 32, subjects aa, al, av, aw, and ay achieved 99.33(±0.55), 99.67(±0.27), 97.32(±1.56), 94.53(±0.42), and 96.20(±1.95), respectively. Accuracy declined slightly for batch size 64 and further for batch size 128. Similar trends were observed in subjects a to g and A01–A09, with the highest accuracy generally observed at batch size 32. Overall, the proposed model performed best with smaller batch sizes. A similar trend was observed for DLRCSPRNN, where six subjects (aa, al, av, aw, ay, and A01) achieved their highest accuracy with batch size 32. However, five subjects (ay, g, A08, A02, and A04) performed best with batch size 128. Across all batch sizes, SD values exceeded 6.20, indicating performance variability. For instance, accuracy for subject av dropped from 98.44(±0.80) at batch size 32 to 77.56(±6.20) at batch size 64 and 73.45(±9.46) at batch size 128.

**Table 2. 5 fold cross validation results for the proposed DLRCSPNN based model.**

| Subject | Original channels (OC) | Retained Channels (RC) | CRR | Accuracy (%) | | |
|---|---|---|---|---|---|---|
| | | | | Batch size:32 | Batch size:64 | Batch size:128 |
| aa | 118 | 49 | 0.41 | 99.33±0.55 | 99.10±0.57 | 97.31±1.31 |
| al | 118 | 93 | 0.78 | 99.67±0.27 | 99.44±0.35 | 98.77±0.82 |
| av | 118 | 47 | 0.39 | 97.32±1.56 | 96.32±2.30 | 91.29±2.15 |
| aw | 118 | 27 | 0.22 | 94.53±0.42 | 91.63±1.80 | 87.95±1.36 |
| ay | 118 | 27 | 0.22 | 96.20±1.95 | 94.42±2.14 | 89.62±0.77 |
| a | 59 | 37 | 0.61 | 99.22±0.70 | 98.54±1.05 | 98.68±1 |
| b | 59 | 31 | 0.52 | 100±0.00 | 99.84±0.31 | 99.69±0.38 |
| c | 59 | 31 | 0.52 | 99.84±0.31 | 99.69±0.62 | 99.06±0.58 |
| d | 59 | 22 | 0.37 | 99.69±0.38 | 98.12±1.17 | 96..56±0.80 |
| e | 59 | 26 | 0.44 | 99.53±0.62 | 99.53±0.38 | 99.38±0.58 |
| f | 59 | 40 | 0.67 | 99.38±0.91 | 97.03±1.81 | 90.16±2.35 |
| g | 59 | 22 | 0.37 | 100±0.00 | 99.69±0.39 | 96.54±0.63 |
| A01 | 22 | 8 | 0.36 | 94.22±0.07 | 91.25±0.09 | 90.70±0.1 |
| A02 | 22 | 6 | 0.27 | 92.08±0.15 | 89.14±0.19 | 84.55±0.22 |
| A03 | 22 | 11 | 0.5 | 92.61±0.18 | 90.79±0.18 | 85.25±0.25 |
| A04 | 22 | 12 | 0.54 | 90.69±0.17 | 76.78±0.25 | 74.31±0.24 |
| A05 | 22 | 6 | 0.27 | 90.78±22.97 | 89.03±0.19 | 86.48±0.21 |
| A06 | 22 | 5 | 0.22 | 90.03±0.17 | 87.80±0.20 | 81.70±0.23 |
| A07 | 22 | 11 | 0.5 | 90.05±0.17 | 87.01±0.19 | 79.02±0.27 |
| A08 | 22 | 11 | 0.5 | 91.96±0.15 | 84.52±0.21 | 78.28±0.23 |
| A09 | 22 | 8 | 0.36 | 93.05±0.14 | 89±0.19 | 81.46±0.24 |

**Table 3. 5 fold cross validation results for the proposed DLRCSPRNN based model.**

| subject | Original channels (OC) | Retained Channels (RC) | CRR | Accuracy (%) | | |
|---|---|---|---|---|---|---|
| | | | | Batch size:32 | Batch size:64 | Batch size:128 |
| aa | 118 | 49 | 0.41 | 98.44±0.38 | 73.91±19.85 | 87.19±14.36 |
| al | 118 | 93 | 0.78 | 97.66±1.15 | 83.27±11.98 | 84.93±7.77 |
| av | 118 | 47 | 0.39 | 98.44±0.80 | 77.56±6.20 | 73.45±9.46 |
| aw | 118 | 27 | 0.22 | 96.09±1.36 | 67.63±10.44 | 69.07±11.66 |
| ay | 118 | 27 | 0.22 | 93.75±1.48 | 76.89±11.58 | 83.27±5.79 |
| a | 59 | 37 | 0.61 | 95.31±2.49 | 75.21±13.36 | 77.85±9.75 |
| b | 59 | 31 | 0.52 | 90.62±2.63 | 73.75±11.56 | 70±10.13 |
| c | 59 | 31 | 0.52 | 84.92±4.14 | 73.12±10.28 | 76.41±6.20 |
| d | 59 | 22 | 0.37 | 90.56±22.15 | 82.50±13.64 | 75.31±17.50 |
| e | 59 | 26 | 0.44 | 73.89 ±7.42 | 67.81±14.13 | 68.91±11.16 |
| f | 59 | 40 | 0.67 | 74.44±8.77 | 73.75±12.02 | 75.62±10.45 |
| g | 59 | 22 | 0.37 | 65.56±3.21 | 67.50±8.16 | 74.69±11.36 |
| A01 | 22 | 8 | 0.36 | 93.78±0.1 | 92.84±0.07 | 87.55±0.14 |
| A02 | 22 | 6 | 0.27 | 93.78±0.1 | 88.68±0.10 | 85.02±0.11 |
| A03 | 22 | 11 | 0.5 | 87.30±0.12 | 85.56±0.06 | 84.77±0.05 |
| A04 | 22 | 12 | 0.54 | 90.05±0.12 | 89.34±0.1 | 83.97±0.12 |
| A05 | 22 | 6 | 0.27 | 84.15±0.06 | 80.9±0.04 | 80.59±0.06 |
| A06 | 22 | 5 | 0.22 | 79.62±0.03 | 78.83±0.06 | 76.43±0.04 |
| A07 | 22 | 11 | 0.5 | 83.93±0.03 | 82.30±0.03 | 80.33±0.06 |
| A08 | 22 | 11 | 0.5 | 81.97±0.04 | 79.58±0.02 | 80.24±0.03 |
| A09 | 22 | 8 | 0.36 | 83.94±0.03 | 81.27±0.12 | 76.21±0.11 |

Figs 4 and 5 provide a batch wise accuracy comparison of DLRCSPRNN and the proposed model (DLRCSPNN) alongside the number of retained channels and CRRs. Figs 6 and 7 illustrate the fold-wise accuracy comparison for DLRCSPRNN and the proposed model for batch size 32 alongside RC. Accuracy varies significantly between subjects, with some achieving consistently high performance across all folds, while others show greater variability. Subjects such as aa, al, av, and A01–A04 maintain accuracy above 90%, whereas a, b, c, d, e, f, and g exhibit lower accuracy, in some cases dropping below 70%.

Sensitivity, also known as recall, hit rate, or true positive rate, refers to the classifier's ability to correctly identify patients from healthy subjects. Fig 8 compares the sensitivity values for DLRCSPRNN and the proposed model at a batch size of 32, showing that DLRCSPRNN has the lowest 5-fold sensitivity for batch 32. Specificity, also known as selectivity or true negative rate, measures the ability of a test to accurately distinguish healthy individuals from patients. High specificity is essential as it ensures that patients are correctly identified and not misclassified as healthy. Fig 9 compares the specificity performance of the two models at a batch size of 32, where the proposed model demonstrates significantly higher specificity compared to the DLRCSPRNN model.

Precision, also referred to as positive predictive value, represents the proportion of true patients within the group identified as patients by the model. Precision is crucial for validating the results' accuracy. Fig 10 illustrates the 5-fold precision comparison between the two models at batch size 32, revealing that DLRCSPRNN exhibits the lowest precision. The proposed model's higher precision suggests that it misclassifies fewer MI tasks. The F1 score, which combines precision and recall, provides a comprehensive performance metric for detecting MI activities. It is calculated as the harmonic mean of precision and recall. Fig 11 presents the F1 score comparison for both the experiments conducted in this study.

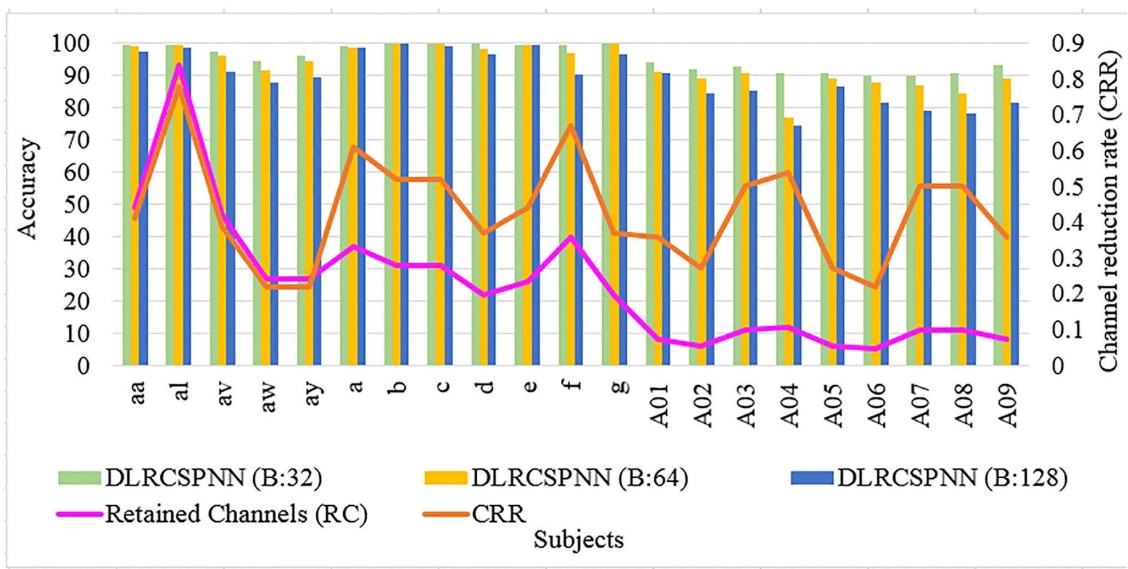

**Fig 4. Batch-wise accuracy comparison of the DLRCSPNN model.**

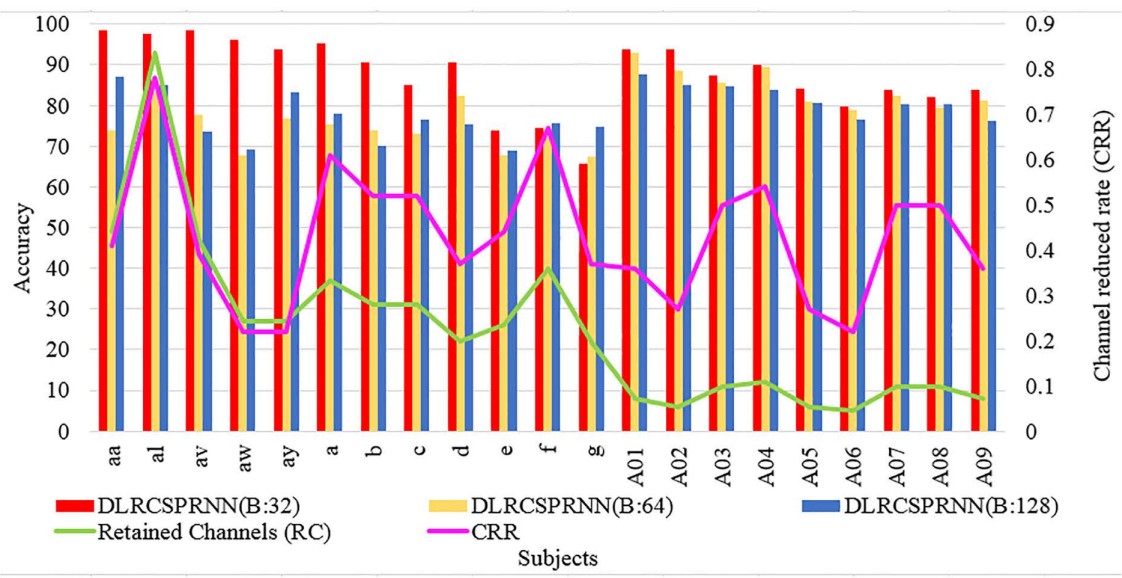

**Fig 5. Batch-wise accuracy comparison of the DLRCSPRNN model.**

Additionally, Figs 12a and 13 shows the ROC curve for both the models of all datasets. A higher ROC generally indicates a better performing model. To interpret ROC, a value of 0.5 indicates no discrimination (random), while a value of 1.0 indicates perfect discrimination between classes. For several subjects (like a, b, d, g, f, al, aw, ay etc.), DLRCSPNN shows a higher score compared to DLRCSPRNN. Overall, The DLRCSPNN model generally outperforms DLRCSPRNN across the subjects. Besides accuracy, we evaluated model execution time. Fig 14 compares execution times, showing that DLRCSPRNN generally takes longer than DLRCSPNN. Notably, processing time was significantly higher for subjects

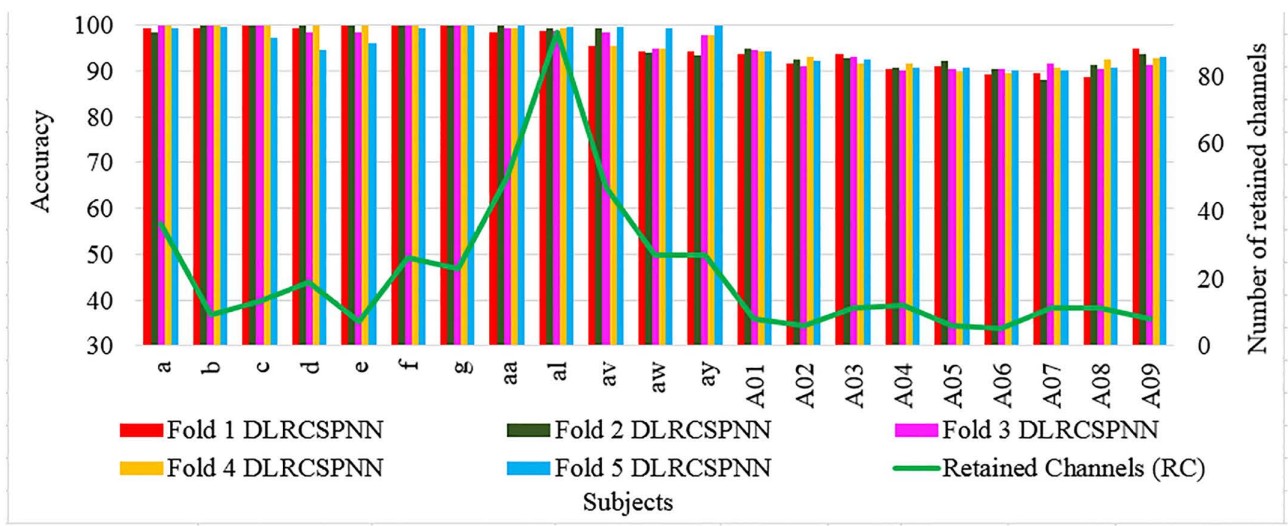

**Fig 6. Fold-wise accuracy comparison of the DLRCSPNN experimented models based on Retained Channels.**

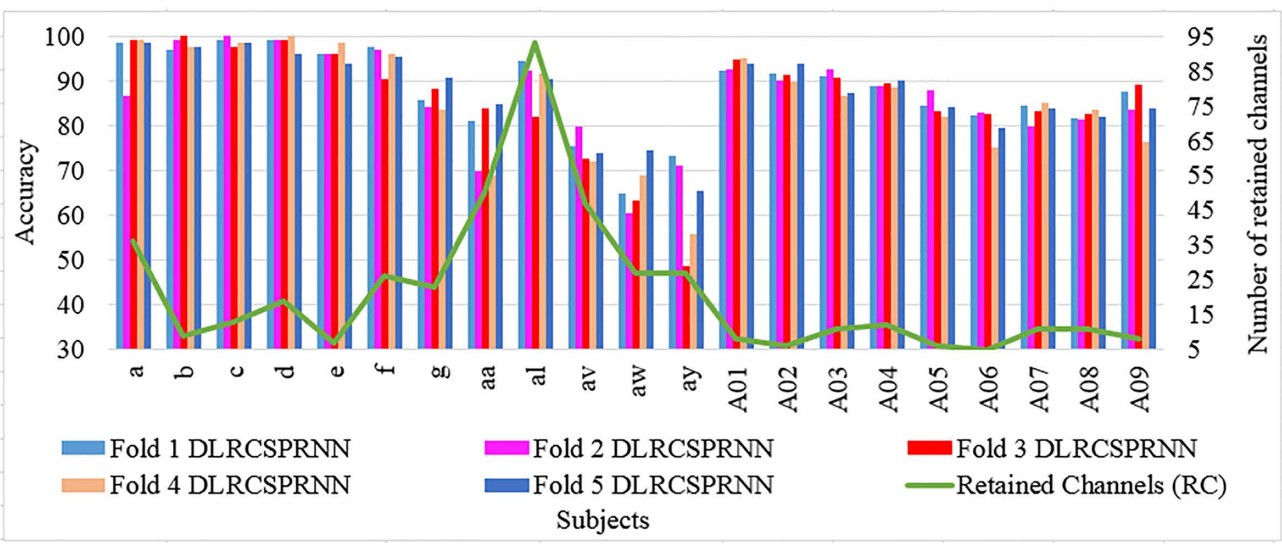

**Fig 7. Fold-wise accuracy comparison of the DLRCSPRNN experimented models based on Retained Channels.**

aw (94.25s vs. 21.51s), g (77.59s vs. 40.93s), and aa (75.92s vs. 49.28s). DLRCSPNN exhibited faster processing, particularly in A02, A04, and A08, suggesting lower computational complexity. For comparison purposes of the two models accuracies, we also applied paired t-test. The result of the paired t-test reveals a T-statistic of 3.84 and a p-value of 0.001, indicating a statistically significant difference. Thus, DLRCSPNN outperforms DLRCSPRNN in both accuracy and model execution time.

Table 4 shows the architectural comparison of the two models. From the table, RNN has six hidden layers with 32,814 trainable parameters and 65,630 optimizer parameters, whereas the proposed model has six hidden layers with 14,382 trainable parameters and 28,766 optimizer parameters, demonstrating lower complexity.

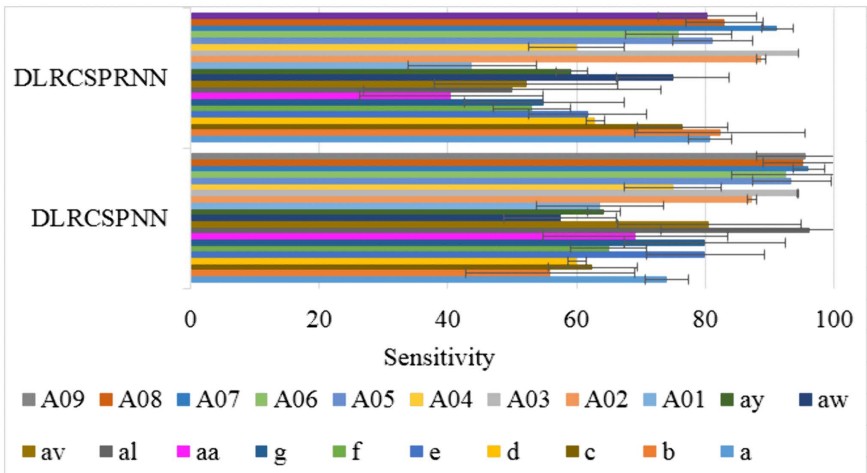

**Fig 8. Sensitivity comparison of the batch size 32 of the experimented models.**

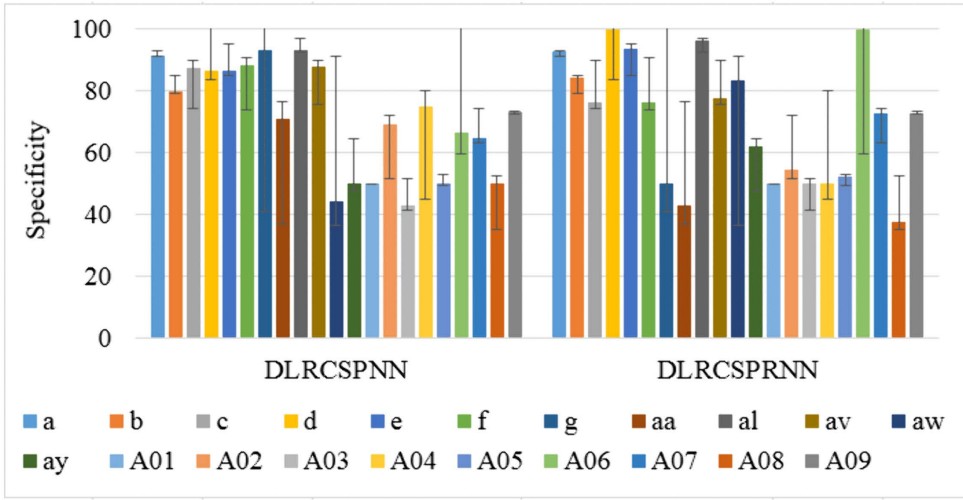

**Fig 9. Specificity comparison of the batch size 32 of the experimented models.**

Through our analysis of the two deep learning models, we found the NN algorithm to be highly efficient. To validate these findings, we conducted an additional experiment using CSP for feature extraction and NN as the classifier to evaluate MI task performance. This was done using 5-fold cross-validation, a batch size of 32, and datasets 1 and 2 with the same reference channels (RC) and channel reduction ratio (CRR). Both frameworks were compared based on data from Tables 2–5 and Figs 4–14.The results consistently showed that DLRCSPNN outperforms CSPNN in classification accuracy. DLRCSPNN achieved accuracy scores ranging from 90% to 100%, while CSPNN's accuracy varied from 62.5% to 98.2%, demonstrating DLRCSPNN's superior classification performance overall. Both models displayed strong specificity, with CSPNN often nearing 100% specificity for some subjects, making it effective in identifying true negatives. DLRCSPNN, though slightly more variable, maintained specificity consistently above 67%. Precision scores revealed a noticeable difference, with CSPNN achieving high values across subjects, whereas DLRCSPNN showed more stable precision

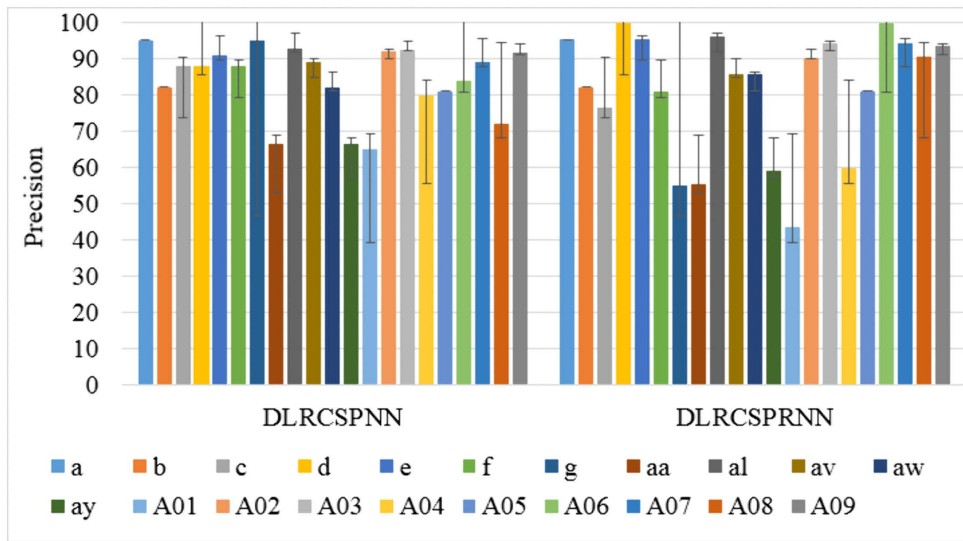

**Fig 10. Precision comparison of the batch size 32 of the experimented models.**

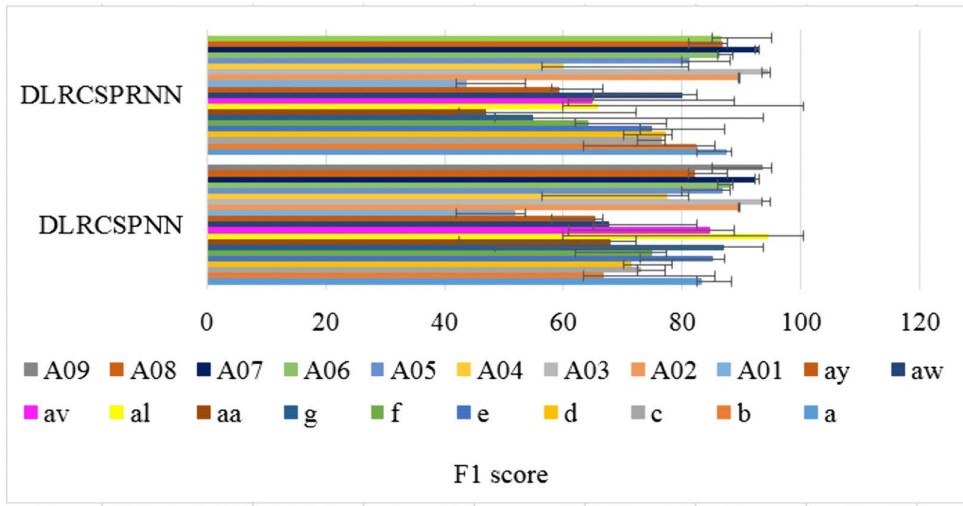

**Fig 11. F1 score comparison of the batch size 32 of the experimented models.**

results, generally staying at or above 65%. The F1 score, which balances precision and recall, also favored DLRCSPNN. Its F1 scores ranged from 65.45 to 94.54, while CSPNN's F1 scores varied over the same range, but with less stability. Execution time was another key differentiator: DLRCSPNN consistently demonstrated faster processing times, ranging from 21.2 to 56.45 seconds, compared to CSPNN's longer times, which ranged from 27.89 to 124.34 seconds. While CSPNN exhibited strong specificity and occasional spikes in sensitivity, DLRCSPNN proved to be more balanced across all performance metrics, including lower execution time. Additionally, we conducted experiments using the original channels for all subjects and compared the results with the DLRCSPNN algorithm across three datasets. As shown in Figs 15 and 16, DLRCSPNN (RC) consistently outperformed DLRCSPNN (OC) in terms of both model execution time and

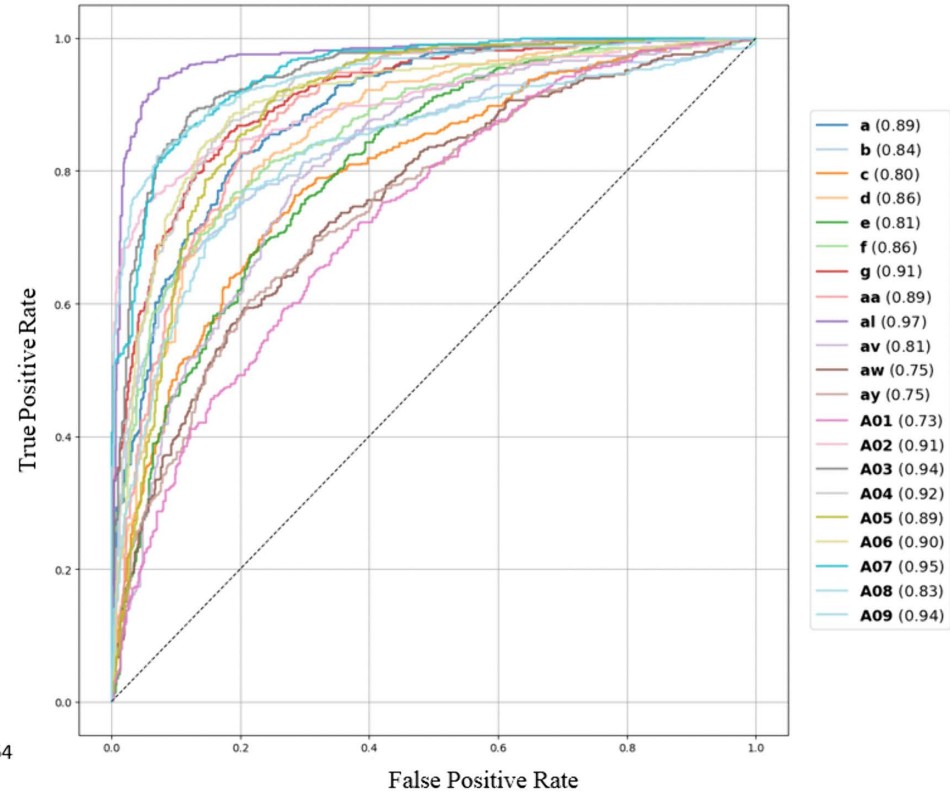

**Fig 12. ROC curve of DLRCSPNN classifiers of 21 subjects of all datasets.**

accuracy. For subjects like 'a', 'b', and 'c', DLRCSPNN (RC) demonstrated significantly faster processing times and higher accuracy. For example, subject 'a' showed 98% accuracy for RC versus 97% for OC, and subject 'b' showed 99% accuracy for RC versus 98% for OC. Even in subjects like 'f', 'g', and 'aa', RC maintained a slight accuracy advantage, such as 97% versus 96% in subject 'g'. In more challenging cases, such as 'aw', 'ay', and A-series subjects, RC continued to be faster while maintaining a marginal edge in accuracy (e.g., 60 seconds for RC versus 65 seconds for OC in 'aw').

These findings suggest that DLRCSPNN (RC) offers a more efficient and reliable model for most subjects, providing a well-balanced trade-off between speed and performance. Therefore, DLRCSPNN (RC) is the preferred choice over DLRCSPNN (OC).

### 3.1. Statistical significance

To statistically validate this difference, we performed a Wilcoxon signed-rank test comparing the two models. The test resulted in a statistic of 50.5 and a p-value of 0.0419, which is below the conventional significance threshold of 0.05. This indicates a statistically significant difference between the full channel set and the retained channel set, suggesting that reducing the number of channels notably impacts the accuracy. Thus, the choice of channels is crucial for the model's performance, and channel reduction has a measurable effect on accuracy.

### 4. Discussion

In this study, we introduced the DLRCSPNN and DLRCSPRNN models for binary MI task classification. This approach is particularly effective at capturing the nonstationary characteristics of EEG data, making it well-suited for classifying MI

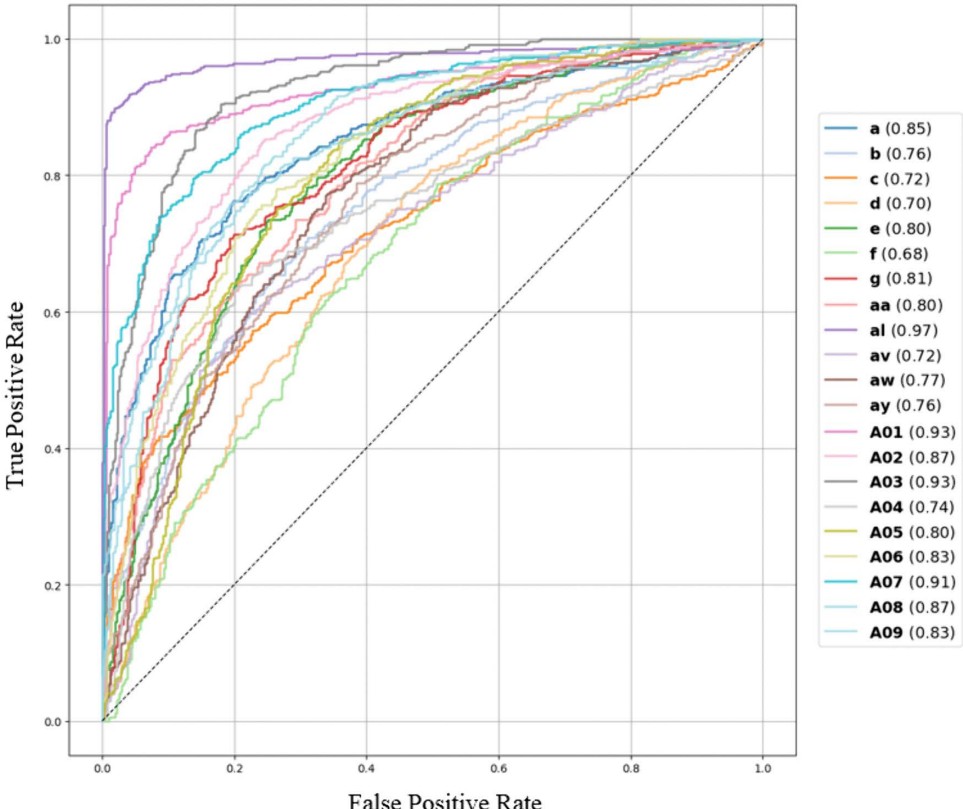

**Fig 13. ROC curve of DLRCSPRNN classifier of 21 subjects of all datasets.**

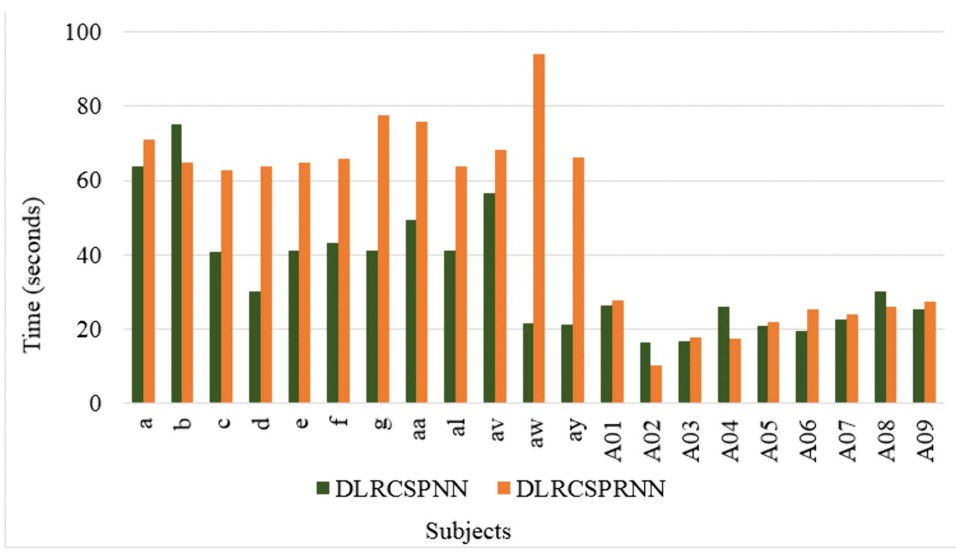

**Fig 14. Time complexity comparison between different classifiers of 21 subjects of all datasets for batch size 32.**

**Table 4. Architectural comparison of the different models.**

| | Layer | Trainable parameters | Total parameters | Optimizer parameters |
|---|---|---|---|---|
| RNN | 2 LSTM layers, 2 Dense layers and 2 Dropout layers | 32,814 | 98,444 | 65,630 |
| NN | 3 Dense layers, 1 Output Dense layer and 2 Dropout layers | 14,382 | 43,148 | 28,766 |

**Table 5. 5 fold cross validation results of different matrices for the CSPNN based model.**

| Sub | OC | RC | CRR | Accuracy (%) | Specificity | Sensitivity | Precision | F1 score | Time(sec) |
|---|---|---|---|---|---|---|---|---|---|
| | | | | Batch size:32 | | | | | |
| aa | 118 | 49 | 0.41 | 69.6 | 61.5 | 76.7 | 69.7 | 73 | 34.17 |
| al | 118 | 93 | 0.78 | 98.2 | 100 | 96.55 | 100 | 98.26 | 59.33 |
| av | 118 | 47 | 0.39 | 62.5 | 60.7 | 64.3 | 62.1 | 63.2 | 31.286 |
| aw | 118 | 27 | 0.22 | 91.1 | 87 | 96 | 86 | 91 | 124.34 |
| ay | 118 | 27 | 0.22 | 94.6 | 96.6 | 92.6 | 93.3 | 94.9 | 91.432 |
| a | 59 | 37 | 0.61 | 97.5 | 95 | 100 | 95 | 97 | 44.05 |
| b | 59 | 31 | 0.52 | 95 | 94 | 95 | 94 | 94 | 95.36 |
| c | 59 | 31 | 0.52 | 92.5 | 87 | 1 | 85 | 92 | 27.89 |
| d | 59 | 22 | 0.37 | 92.5 | 67 | 85 | 87 | 75 | 64.82 |
| e | 59 | 26 | 0.44 | 97.5 | 100 | 94 | 100 | 97 | 78.20 |
| f | 59 | 40 | 0.67 | 95 | 89 | 100 | 91 | 95 | 40.81 |
| g | 59 | 22 | 0.37 | 87.8 | 91 | 82 | 87.5 | 84.8 | 33.11 |

tasks in BCI applications. Our findings demonstrate that the DLRCSPNN model consistently outperforms other models, including CSPNN and DLRCSPRNN, across various performance metrics. When compared to existing channel selection methods, DLRCSPNN achieves superior classification accuracy. For Dataset 1, it improved accuracy by 3.27% to 42.53% over previous methods, while for Dataset 2 and 3, it achieved a 5% to 45% and 1% to 17.47% improvement respectively. A key finding, supported by Figs 15 and 16, is that DLRCSPNN with retained channels (RC) not only achieved comparable or superior accuracy to the full-channel configuration (OC) but also demonstrated significantly faster model execution times. For instance, subject *a* achieved 98% accuracy using RC compared to 97% with OC, while reducing computation time from 78 seconds to 64 seconds. The Wilcoxon signed-rank test (statistic = 50.5, $p$ = 0.0419) confirmed that this improvement in efficiency is statistically significant. To validate the effectiveness of our channel selection approach, Figs 17–19 compare DLRCSPNN with existing state-of-the-art channel selection methods across three benchmark datasets. The three bar charts compare the classification accuracy across subjects using various methods, with accuracy values represented by bar heights and the number of EEG channels retained indicated on top of each bar. In all three datasets, DLRCSPNN maintains high classification accuracy when utilizing RC compared to alternative approaches. For dataset 1, DLRCSPNN outperforms or matches state-of-the-art approaches in accuracy across most cases. For instance, in subject aa, DLRCSPNN achieves an impressive 99.84% accuracy with 49 channels, surpassing RCSPA-RSVM (Regularized Common Spatial Pattern with Aggregation- Regularized Support Vector Machine) algorithm in Tiwari and Chaturvedi (89.34%) and PCC-CSP (Pearson's correlation coefficient-CSP) method in Gaur et al (75.89%) [33,51]. Similarly, in subject al, DLRCSPNN maintains 99.69% accuracy with 93 channels, while other methods, such as BHS-CSP (binary harmony search-CSP) and BFN-SL (brain function networks -Synchronization Likelihood) developed by Shi et al and Du et al, respectively fall below 82% [39,52]. In several cases, DLRCSPNN achieves near-perfect classification, such as in subject ay where it reaches100% accuracy, demonstrating its reliability. For dataset 2, DLRCSPNN again demonstrates high accuracy while while using fewer channels. For subject a, it achieves 99.33% accuracy with just 37 channels, outperforming BHS-CSP (62.5%) and BFN-SL (85.3%) frameworks. Similarly, for b, DLRCSPNN achieved 99.67%

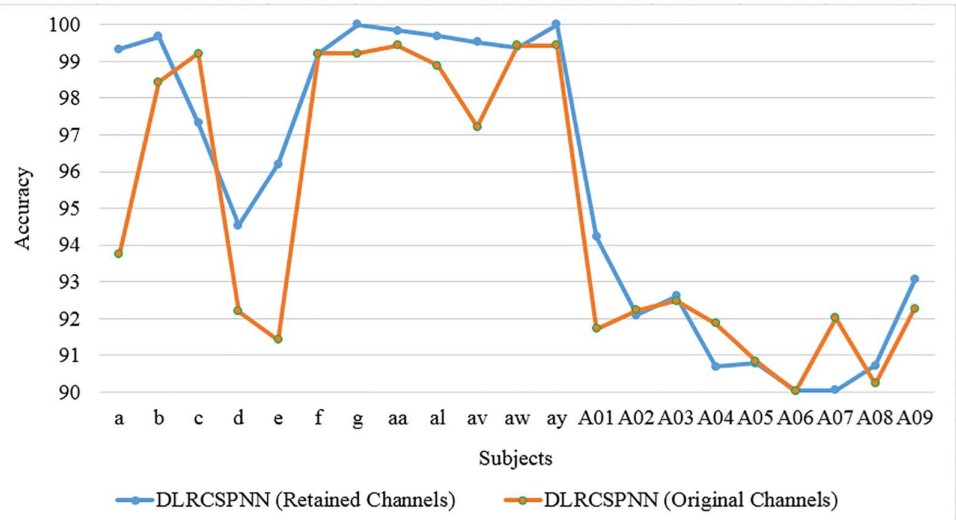

**Fig 15. Accuracy comparison between DLRCSPNN models using retained channels (RC) and original channels (OC) for all subjects.**

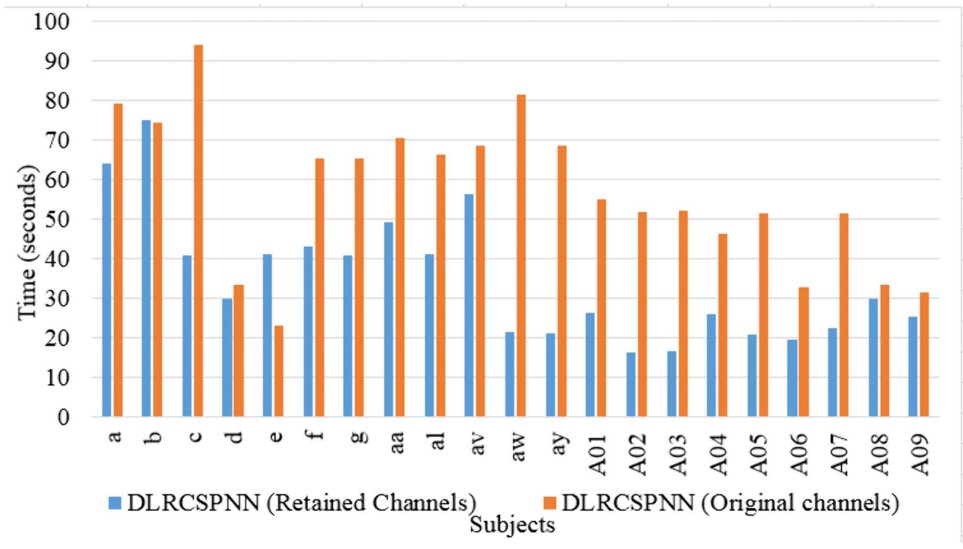

**Fig 16. Computational time comparison of DLRCSPNN model between retained channels (RC) and original channels (OC) for all the subjects.**

accuracy with only 31 channels, surpassing methods like NMI-HOG (normalized mutual information- Histogram of oriented gradient) and CDCS-CSP introduced by Tang et al and Qin et al which fall short at 63% [28,53]. Across other subjects, DLRCSPNN continues to outperform competing methods. In dataset 3, DLRCSPNN again proves to be highly efficient in achieving high accuracy while using fewer channels. In A02, it reaches 92.08% accuracy with just 6 channels, whereas DGAFF-TSCNN algorithm developed by Khalid et al achieved only 73.41% accuracy with the same number of channels [26]. A similar pattern is seen in A03, where Khalid et al obtain slightly higher accuracy (97.82%) with fewer channels [7], while DLRCSPNN achieved 92.61% with 11 channels. In cases like A08, LMSST-LDA developed by Dovedi et al achieved the highest accuracy (99.65%) using 10 channels, DLRCSPNN maintains a competitive 91.96% accuracy with

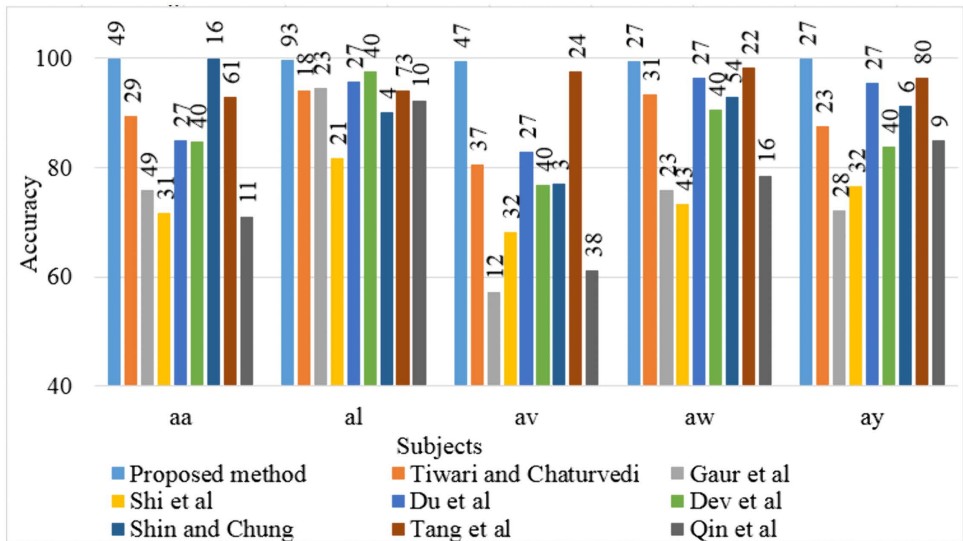

**Fig 17. Comparison of classification accuracy of our developed method and existing channel selection methods of dataset 1.** Accuracy values are shown as bar heights, while the numeric values on top of each bar represent the number of EEG channels retained after channel selection.

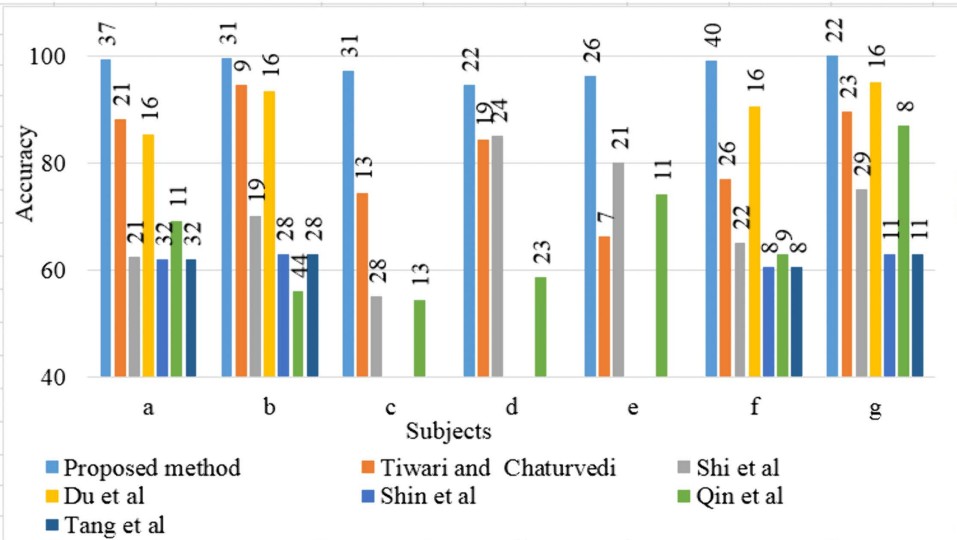

**Fig 18. Comparison of classification accuracy of our developed method and existing channel selection methods of dataset 2.** Accuracy values are shown as bar heights, while the numeric values on top of each bar represent the number of EEG channels retained after channel selection.

11 channels [30]. In a few instances, such as subjects A03 or A08, our method exhibited slightly lower accuracy compared to specific high-performing baselines. We attribute this to our model's aggressive pruning, which, while highly effective for reducing computational overhead, may occasionally sacrifice marginal accuracy in favor of generalizability and resource efficiency. Additionally, EEG signals are inherently subject-specific and nonstationary, which can influence per-subject performance under a reduced-channel setting. Overall, the results across all three datasets highlight the effectiveness of DLRCSPNN in balancing accuracy and model execution time. It consistently outperforms or matches the best-performing

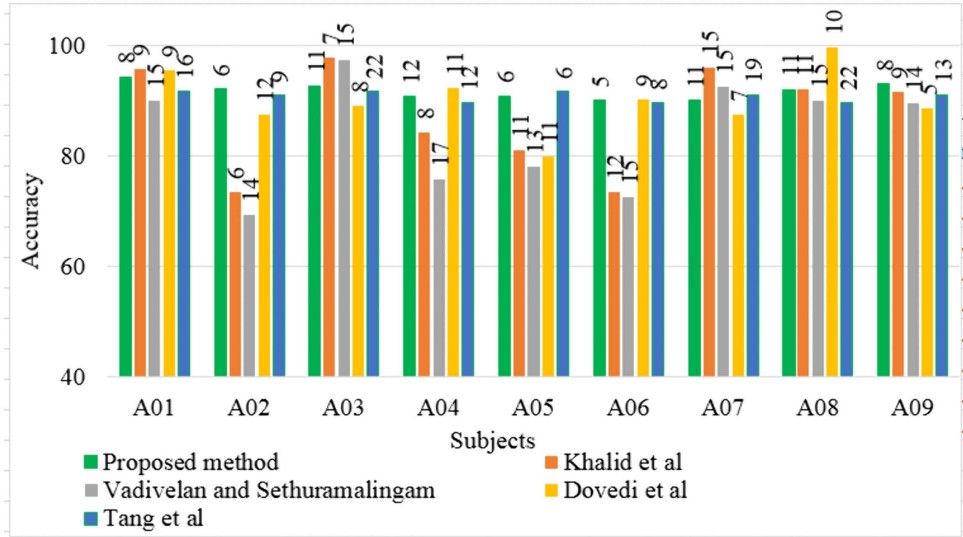

**Fig 19. Comparison of classification accuracy of our developed method and existing channel selection methods of dataset 3.** Accuracy values are shown as bar heights, while the numeric values on top of each bar represent the number of EEG channels retained after channel selection.

state-of-the-art algorithms, demonstrating its robustness in EEG-based MI tasks classification. The model's ability to achieve high accuracy with fewer channels makes it a strong candidate for real-world BCI applications, where reducing hardware complexity and computational load is essential

## 5. Conclusion

In this article, we introduced a DLRCSPNN-based channel selection algorithm aimed at analyzing large-scale brain signal data for advanced BCI technology. Our method involved selecting channels based on their correlation coefficients, discarding those with a value below 0.5. To validate the effectiveness of this approach, we conducted extensive experiments using 5-fold cross-validation. For Dataset 1, our algorithm consistently outperformed all other approaches, achieving the highest accuracy across all subjects, with improvements ranging from 3.27% to 42.53% in detecting MI tasks for BCI applications. Similarly, in Dataset 2 and 3 our algorithm demonstrated superior accuracy compared to existing methods, with improvements between 5% to 45% and 1% to 17.47% respectively. These results highlight the novelty and effectiveness of our approach, offering a new perspective on EEG data analysis and contributing to advancements in BCI technology. The high classification accuracy achieved by our framework suggests that even short EEG data segments of just 3 seconds are sufficient for identifying MI tasks or movements. Consequently, this framework will provide valuable insights for technologists in developing software or apps aimed at improving the lives of motor-disabled individuals. Future research will involve evaluating the system's performance using a hybrid deep learning method to achieve higher performance in EEG data analysis.

## Author contributions

**Conceptualization:** Taslima Khanam.

**Formal analysis:** Taslima Khanam.

**Investigation:** Taslima Khanam.

**Methodology:** Taslima Khanam, Kabir Ahmad.

**Supervision:** Siuly Siuly, Hua Wang.

**Writing – original draft:** Taslima Khanam.

**Writing – review & editing:** Taslima Khanam, Siuly Siuly, Kabir Ahmad, Hua Wang.

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
