## [Decision Letter · Decision Letter 0]

28 Nov 2024

Dear Dr. Khanam,

Thank you for submitting your manuscript to PLOS ONE. After careful consideration, we feel that it has merit but does not fully meet PLOS ONE’s publication criteria as it currently stands. Therefore, we invite you to submit a revised version of the manuscript that addresses the points raised during the review process.

The paper has been reviewed by two reviewers, and both have shown serious concerns regarding the methodology. The authors are encouraged to address the comments and resubmit the manuscript.

We look forward to receiving your revised manuscript.

Kind regards,

Noman Naseer, PhD

Academic Editor

PLOS ONE

Journal Requirements:

3. In the online submission form, you indicated that your data is available only on request from a third party. Please note that your Data Availability Statement is currently missing the contact details for the third party, such as an email address or a link to where data requests can be made. Please update your statement with the missing information.

Additional Editor Comments:

The paper has been reviewed by two reviewers, and both have shown serious concerns regarding the methodology. The authors are encouraged to address the comments and resubmit the manuscript.

Reviewers' comments:

Reviewer's Responses to Questions

**Comments to the Author**

1. Is the manuscript technically sound, and do the data support the conclusions?

Reviewer #1: Partly

Reviewer #2: No

2. Has the statistical analysis been performed appropriately and rigorously?

Reviewer #1: No

Reviewer #2: No

3. Have the authors made all data underlying the findings in their manuscript fully available?

Reviewer #1: Yes

Reviewer #2: No

4. Is the manuscript presented in an intelligible fashion and written in standard English?

Reviewer #1: Yes

Reviewer #2: No

Reviewer #1: The manuscript proposed an AI-based channel selection algorithms that are both fast and accurate for identifying EEG-based motor imagery (MI) movements. A novel hybrid method combining statistical tests with Bonferroni correction is introduced to achieve an effective channel reduction rate for selecting MI task-related EEG channels and an automatic deep learning-based framework for MI task classification. The paper is well written and structured, however it needs improvement.

Following are the suggestions to improve the quality of paper:

1. Font size and style is different at some places. It should be same in the text, paragraph, figure and tables. Follow the journal template and guidelines.

2. Authors have mentioned that proposed algorithm is tested and validated for two EEG datasets of 5 and 7 participants, respectively. However, still the sample of overall 12 participant is too small to conclude results and claims. There are open access datasets of EEG available on the internet with adequate sample size and number of participants. The proposed algorithm should be tested and validated on larger datasets to strengthen the claims and contributions of the study.

3. In the Table 2: 5 fold average performance results for the proposed DLRCSPNN based model, there are multiple accuracies for participants showing 100% results; this may be due to over-fitting of the classifier. This need to be verified for the results of DLRCSPRNN results as well.

4. Student’s t-test and Bonferroni correction is applied to select the channels. Is there any significance statistical analysis applied on the results; accuracies?

5. In Table 4. Architectural comparison of the different models are discussed, however the hardware details are not described on which both the algorithms are computed.

Reviewer #2: This paper, in its current form, lacks the essential elements of a well-structured and scientifically rigorous study, making it difficult to consider it a strong contribution to the field. Significant issues across multiple sections need to be addressed:

1. Introduction and Literature Review:

The introduction section is missing a detailed literature review, particularly for the classification methods discussed. A comprehensive background and context are essential to highlight the novelty and relevance of the study.

2. Figure Quality:

o Figure 1: The image is blurry and appears AI-generated. Its quality needs significant improvement to enhance clarity and readability.

o Figures 2 and 3: These figures are also blurry and lack detailed representation. They need to be redrawn with higher resolution and properly annotated details.

3. Dataset Details:

o Clarify the distinction between Dataset 1 (imaginary motor tasks involving the right hand and foot) and Dataset 2 (motor acquisition).

o Provide justification for why two participants performed left-hand and foot activities, while others focused on the right-hand tasks.

o Discuss the implications of such variations in task allocation on the results.

4. Methodological Concerns:

o Line 151: The sentence mentioning "might lose important information in discarded channels" raises concerns. Results should be compared with all channels to confirm whether the channel selection method inadvertently excludes useful information.

o Line 198: Correct the typo where "=1=1" is mentioned.

o Details on the DLRCSP algorithm are missing. Explain how the algorithm is designed and applied in the study.

5. Neural Network Representation:

o Figure 3: The representation is unclear. It suggests input data is fed to all three dense layers separately, which is logically inconsistent. Recheck the flow of information within the network layers and clarify the purpose of lines joining the blocks.

6. Experimental Setup and Results:

o Line 238: Merely training the model on k−1k-1k−1 subsets (with only 42 samples) is insufficient. Justify how the training data is adequate for robust model performance.

o When comparing the proposed model with RNN algorithms, ensure the number of neurons and sequence of layers are consistent. Differences in network architecture can significantly impact results and invalidate comparisons.

o Results are only compared with a single RNN algorithm (LSTM). Provide a rationale for choosing RNN and LSTM and explain why comparisons with other architectures (e.g., GRU) are omitted.

7. Subject-Specific Analysis:

o Figure 7: Results for subject "ay" are missing when using RNN. Ensure all subjects are included in the analysis for consistency.

o Comparisons between datasets (motor imagery vs. motor acquisition) are problematic as the datasets inherently differ in nature, which could skew results.

8. Comparison and Benchmarking:

o Comparative results are not adequately presented in tables or bar charts. Include comparisons with benchmark algorithms or methodologies to demonstrate the performance of the proposed approach.

o Ensure all bar charts include comparative results for clarity.

9. Discussion Section:

o The discussion section does not adequately support the results or compare them with findings from previous studies. Expand this section to include critical analysis and comparisons with related works.

10. Methodology and Results Flow:

• Experimental setups mentioned in the results section should be moved to the methodology section for better structure and flow.

**Do you want your identity to be public for this peer review?** For information about this choice, including consent withdrawal, please see our Privacy Policy

Reviewer #1: No

Reviewer #2: No

---

## [Author Response · Author response to Decision Letter 1]

29 Mar 2025

We would like to thank the editor and all reviewers for their time, constructive comments and useful suggestions which have led to a much-improved manuscript. To clearly address the amendments performed in the revised manuscript, we have quoted the specific reviewers’ comments in our responses correspondingly. In Bold are the comments from editor/reviewers, the plain blue colour text is our direct responses and the Italic is actual changes implemented, i.e. what we added, deleted, rewrote or modified. All changes to the original version of the document are now in red ink to facilitate the review process.

Reviewer #1: The manuscript proposed an AI-based channel selection algorithms that are both fast and accurate for identifying EEG-based motor imagery (MI) movements. A novel hybrid method combining statistical tests with Bonferroni correction is introduced to achieve an effective channel reduction rate for selecting MI task-related EEG channels and an automatic deep learning-based framework for MI task classification. The paper is well written and structured, however it needs improvement. Following are the suggestions to improve the quality of paper:

1. Font size and style is different at some places. It should be same in the text, paragraph, figure and tables. Follow the journal template and guidelines.

Answer: Thank you for your valuable feedback. We sincerely apologize for the inconsistency in font size and style. We have carefully revised the manuscript to ensure uniformity in text, paragraphs, figures, and tables, adhering strictly to the journal’s template and guidelines. We appreciate your attention to detail and your efforts in improving the quality of our work.

2. Authors have mentioned that proposed algorithm is tested and validated for two EEG datasets of5 and 7 participants, respectively. However, still the sample of overall 12 participant is too small to conclude results and claims. There are open access datasets of EEG available on the internet with adequate sample size and number of participants. The proposed algorithm should be tested and validated on larger datasets to strengthen the claims and contributions of the study.

Answer: Thank you very much for your thoughtful comments and valuable feedback. We appreciate your concern regarding the sample size in our study. We acknowledge that a larger sample size could indeed strengthen the validity of the results and the generalizability of the findings. In response to your comment, we have now included an additional widely recognized and publicly available dataset, BCI Competition IVa Dataset 2a, which consists of data from 9 participants. This dataset is commonly used in the EEG research community and will help enhance the robustness of our experiment. We hope that this addition addresses your concern and improves the overall quality of our study.

3. In the Table 2: 5 fold average performance results for the proposed DLRCSPNN based model, there are multiple accuracies for participants showing 100% results; this may be due to over-fitting of the classifier. This need to be verified for the results of DLRCSPRNN results as well.

Answer: Thank you for your insightful comment regarding the potential overfitting observed in the accuracy results. To address this, we implemented several strategies to ensure the robustness of the models. First, we employed 5-fold cross-validation, ensuring that each data split was used for validation exactly once, which helps in reducing variance and improving the model's generalization. To further mitigate overfitting, we used dropout layers (0.5 rate) in both the Neural Network (NN) and Recurrent Neural Network (RNN) classifiers. These steps, combined with data standardization using StandardScaler and weight regularization with the Adam optimizer, ensured that the models would not memorize the training data but instead generalize better. We also tracked performance using standard deviation calculations, ROC curves, and AUC scores to detect any signs of overfitting. By training the models for 100 epochs and testing with various batch sizes, we further ensured that the results were not a result of overfitting. The experiments were performed on Google Colab with 2 CPU cores and 12.67 GB of RAM, without GPU acceleration, providing a consistent computational environment. We believe these measures successfully addressed overfitting and contributed to the reliable results we presented. Improvement made in the manuscript page no. 14-17.

4. Student’s t-test and Bonferroni correction is applied to select the channels. Is there any significance statistical analysis applied on the results; accuracies?

Answer: Thank you for your valuable feedback. Regarding the significance of the statistical analysis applied to the results, we utilized the paired t-test for comparison of the DLRCSPNN and DLRCSPRNN models’ accuracies. The results of the paired t-test revealed a T-statistic of 3.84 and a p-value of 0.001, indicating a statistically significant difference between the models. Additionally, we applied Wilcoxon signed-rank test to compare the DLRCSPNN (RC) model with the DLRCSPNN (OC) model, based on the reduced and original channel sets, respectively. This analysis provides further insight into the differences in accuracy between the two models under different conditions. We hope this clarifies the statistical methods used for the analysis of the results and accuracies. Pge no. 22-23, 26

5. In Table 4. Architectural comparison of the different models are discussed, however the hardware details are not described on which both the algorithms are computed.

Answer: Thank you for your insightful comment. Regarding the hardware details on which both algorithms were computed, we used Google Colab, a cloud-based platform, with the following specifications: 2 CPU cores, 12.67 GB of RAM, and no GPU acceleration. The computational environment is virtualized, and the experiments were conducted without dedicated GPU support, which may affect processing time for resource-intensive tasks. We hope this additional context clarifies the hardware setup used for the computations of both algorithms. Page no.16.

Reviewer #2: This paper, in its current form, lacks the essential elements of a well-structured and scientifically rigorous study, making it difficult to consider it a strong contribution to the field. Significant issues across multiple sections need to be addressed:

1. Introduction and Literature Review: The introduction section is missing a detailed literature review, particularly for the classification methods discussed. A comprehensive background and context are essential to highlight the novelty and relevance of the study.

Answer: We greatly appreciate your insightful feedback. In response to your comment, we have addressed the need for a more comprehensive literature review in the introduction, with a particular focus on the classification methods used in the study. The expanded literature review is now detailed on pages 2-4 of the manuscript.

In this section, we have provided an in-depth discussion of key motor imagery task classification methods used in EEG analysis. We specifically explored various channel selection techniques, including genetic algorithms, common spatial patterns, and other methods, outlining both their strengths and limitations. By reviewing these existing approaches, we aim to demonstrate the novelty and relevance of our study, particularly in relation to our AI-based channel selection method, which combines statistical tests with Bonferroni correction to improve classification accuracy while reducing computational complexity.

We believe that this more detailed background helps to clearly position our work within the broader context of BCI technology, and we hope this revision effectively highlights the significance of our contributions.

2. Figure Quality:

o Figure 1: The image is blurry and appears AI-generated. Its quality needs significant improvement to enhance clarity and readability.

o Figures 2 and 3: These figures are also blurry and lack detailed

Answer: We appreciate your observation regarding Figure 1, 2 and 3. We have now improved its quality to ensure better clarity and readability. Page 5, 7, 12.

3. Dataset Details: o Clarify the distinction between Dataset 1 (imaginary motor tasks involving the right hand and foot) and Dataset 2 (motor acquisition).

o Provide justification for why two participants performed left-hand and foot activities, while others focused on the right-hand tasks.

o Discuss the implications of such variations in task allocation on the results.

Answer: Thank you for your valuable feedback. We appreciate your comments and are happy to provide clarification regarding the datasets and task allocation. In the manuscript, page no. 5 to 6.

BCI Competition III Dataset IVa:

This dataset is focused on motor imagery tasks involving the right hand and right foot. Participants were asked to imagine movements of either their right hand (class 1) or right foot (class 2), with no actual physical movement involved. EEG data were recorded using 118 electrodes following the 10/20 international system. The dataset includes data from five participants, each completing 280 trials, with each trial lasting 3.5 seconds. The primary aim of this dataset is to study the brain's response to imagined motor tasks, which can be used in applications such as brain-computer interfaces (BCI). Importantly, no physical movements were performed by the participants, as the focus was solely on motor imagery.

BCI Competition IV Dataset 1:

Unlike Dataset IVa, BCI Competition IV Dataset 1 involves both motor imagery and motor execution tasks. Participants performed tasks involving their left hand, right hand, and foot. EEG data were recorded using 59 channels from seven participants. During the trials, participants either imagined (motor imagery) or physically executed (motor execution) the movements of different body parts. The goal of this dataset is to investigate brain activity during both motor imagery and actual movement, focusing on the differences between these two types of brain activity. This dataset is particularly relevant for real-world applications, such as controlling prosthetics and other BCI systems that require accurate motor command decoding.

Regarding your question about why certain participants performed left-hand and foot activities, while others focused on right-hand tasks, the dataset does not provide explicit reasons for these assignments. From the available documentation, it appears that the task allocation was based on the participants’ choices or experimental design, but further details are not provided by the dataset creators.

BCI Competition IV Dataset 2a:

This dataset involves motor imagery tasks across a broader range of body parts, including the left and right hand, both feet, and the tongue. The dataset includes data from nine participants, with recordings made during two sessions. Each session contains 288 trials, divided into six runs. The first session contains labeled data for model training, while the second session contains unlabeled data for evaluation. EEG signals were recorded using 22 Ag/AgCl electrodes and three EOG channels, sampled at 250 Hz. The focus of this dataset is on analyzing motor imagery for different body parts, making it valuable for developing BCIs that can decode complex motor imagery tasks across a variety of body parts.

Distinctions:

1. BCI Competition III Dataset IVa focuses only on motor imagery tasks involving the right hand and right foot, whereas BCI Competition IV Dataset 1 includes both motor imagery and motor execution tasks involving the left hand, right hand, and foot.

2. BCI Competition IV Dataset 1 uses 59 channels for data collection, while BCI Competition III Dataset IVa uses 118 channels, making Dataset IVa higher in channel count.

3. BCI Competition IV Dataset 2a includes motor imagery tasks for a wider range of body parts (left and right hand, both feet, and tongue) and has a larger number of participants (nine participants), compared to Dataset 1, which uses only seven participants.

In summary, the datasets differ in the type of tasks (motor imagery vs. motor execution), the number of body parts involved, the number of participants, and the number of EEG channels used. We hope this clears up the distinctions between the datasets and the rationale for task allocation.

4. Methodological Concerns:

o Line 151: The sentence mentioning "might lose important information in discarded channels" raises concerns. Results should be compared with all channels to confirm whether the channel selection method inadvertently excludes useful information.

Answer: We appreciate your valuable feedback and the opportunity to address your concern regarding the potential loss of important information when discarding channels. In response, we performed a direct comparison between the full channel set and the selected channel set for the DLRCSPNN model, specifically evaluating both accuracy and computational time in the manuscript page no.24-26.

Additionally, we conducted experiments using the original channels for all subjects and compared the results with the DLRCSPNN algorithm across three datasets. As shown in Figures 12 and 13, DLRCSPNN (RC) consistently outperformed DLRCSPNN (OC) in terms of both time efficiency and accuracy. For subjects like 'a', 'b', and 'c', DLRCSPNN (RC) demonstrated significantly faster processing times and higher accuracy. For example, subject 'a' showed 98% accuracy for RC versus 97% for OC, and subject 'b' showed 99% accuracy for RC versus 98% for OC. Even in subjects like 'f', 'g', and 'aa', RC maintained a slight accuracy advantage, such as 97% versus 96% in subject 'g'. In more challenging cases, such as 'aw', 'ay', and A-series subjects, RC continued to be faster while maintaining a marginal edge in accuracy (e.g., 60 seconds for RC versus 65 seconds for OC in 'aw').

These findings suggest that DLRCSPNN (RC) offers a more efficient and reliable model for most subjects, providing a well-balanced trade-off between speed and performance. Therefore, DLRCSPNN (RC) is the preferred choice over DLRCSPNN (OC). To statistically validate this difference, we performed a Wilcoxon signed-rank test comparing the two models. The test resulted in a statistic of 50.5 and a p-value of 0.0419, which is below the conventional significance threshold of 0.05. This indicates a statistically significant difference between the full channel set of model and the reduced channel set of model, suggesting that reducing the number of channels notably impacts the accuracy.

We believe this additional analysis provides a clear justification for the channel selection method used and assures that the results are reliable and well-supported. Thank you for bringing this to our attention.

o Line 198: Correct the typo where "=1=1" is mentioned.

Answer: We appreciate your attention to detail. The typo "=1=1" in Line 198 has been corrected. Thank you for bringing this to our attention. Page no. 10.

o Details on the DLRCSP algorithm are missing. Explain how the algorithm is designed and applied in the study.

Answer: Thank you for your insightful comment. We have now provided a detailed explanation of the DLRCSP algorithm in the manuscript to address your concern in page no. 9-10.

5. Neural Network Representation:

o Figure 3: The representation is unclear. It suggests input data is fed to all three dense layers separately, which is logically inconsistent. Recheck the flow of information within the network layers and clarify the purpose of lines joining the blocks.

Answer: We appreciate your attention to detail. Figure 3 has been corrected.

6. Experimental Setup and Results:

o Line 238: Merely training the model on k−1k-1k−1 subsets (with only 42 samples) is insufficient. Justify how the training data is adequate for robust model performance.

Answer: Thank you for your constructive comment. We apologize for the oversight regarding the use of only k−1 subsets (with 42 samples) for training the model. After carefully reviewing the methodology, we have made the necessary corrections. We have adjusted the dataset and the training procedure to ensure that the model is trained on an adequate amount

---

## [Decision Letter · Decision Letter 1]

13 Jun 2025

Dear Dr. Khanam,

Thank you for submitting your manuscript to PLOS ONE. After careful consideration, we feel that it has merit but does not fully meet PLOS ONE’s publication criteria as it currently stands. Therefore, we invite you to submit a revised version of the manuscript that addresses the points raised during the review process.

**ACADEMIC EDITOR:** Some minor revisions are still required.

We look forward to receiving your revised manuscript.

Kind regards,

Noman Naseer, PhD

Academic Editor

PLOS ONE

Journal Requirements:

Additional Editor Comments:

Still some revision are required.

Reviewers' comments:

Reviewer's Responses to Questions

**Comments to the Author**

Reviewer #1: (No Response)

Reviewer #2: All comments have been addressed

2. Is the manuscript technically sound, and do the data support the conclusions?

Reviewer #1: Yes

Reviewer #2: Partly

3. Has the statistical analysis been performed appropriately and rigorously?

Reviewer #1: Yes

Reviewer #2: Yes

4. Have the authors made all data underlying the findings in their manuscript fully available?

Reviewer #1: Yes

Reviewer #2: No

5. Is the manuscript presented in an intelligible fashion and written in standard English?

Reviewer #1: Yes

Reviewer #2: Yes

Reviewer #1: The authors have improved the manuscript significantly, however following concerns still need to be addressed.

1. Font Size and style: This still need to addressed, it is recommended to review the complete thoroughly and use same font style in text, tables and figures. It is evident that font style of references is different from other text in the manuscript. Table 5 has multiple font styles.

2. Figure size and aspect ratio: the figure font style and size should be same as text, also keep aspect ratio of figure readable. Moreover, figure 1 text (flowchart) is not readable.

3. Results of Wilcoxon signed-rank test should be included in the manuscript as separate heading “statistical significance”

Reviewer #2: Channel Selection:

The proposed channel reduction method is novel but lacks comparison with existing channel selection techniques.

The paper does not specify which or how many channels were selected from each dataset.

Suggestion: Include comparative results with standard methods and clearly state selected channels per dataset.

Dataset Details:

It’s unclear if all three datasets are binary classification problems.

Suggestion: Clarify class details for each dataset.

Figure 10 Clarity:

The combined plot is unclear.

Suggestion: Plot DLRCSPNN and DLRCSPRNN results separately for better clarity.

Efficiency Claims (Line 550):

Efficiency isn’t quantified in results.

Suggestion: Add computational time comparison with baseline NN and RNN (without DLRCSP). Expand Figure 11 accordingly.

Unsupported Claim (Line 642):

No comparative accuracy results are shown for existing channel selection methods.

Suggestion: Remove or support the claim with data.

Discussion – Lower Accuracy (Figures 12–16):

Other studies show higher ACC and RC.

Suggestion: Discuss possible reasons for lower performance of the proposed method.

**Do you want your identity to be public for this peer review?** For information about this choice, including consent withdrawal, please see our Privacy Policy

Reviewer #1: **Yes: ** Syed Hammad Nazeer Gilani

Reviewer #2: No

---

## [Author Response · Author response to Decision Letter 2]

17 Jul 2025

Responses of Editor and Reviewers’ comments/suggestions

Manuscript title: A novel channel reduction concept to enhance the classification of motor imagery tasks in brain-computer interface systems

Authors: Taslima Khanam¹, Siuly Siuly1, Kabir Ahmad2, and Hua Wang1

We would like to thank the editor and all reviewers for their time, constructive comments and useful suggestions which have led to a much-improved manuscript. To clearly address the amendments performed in the revised manuscript, we have quoted the specific reviewers’ comments in our responses correspondingly. In Bold are the comments from editor/reviewers, the plain blue colour text is our direct responses and the Italic is actual changes implemented, i.e. what we added, deleted, rewrote or modified. All changes to the original version of the document are now in red ink to facilitate the review process.

Response to Reviewer #1:

We thank the reviewer for acknowledging the improvements made and for providing additional constructive feedback. We have carefully addressed each of the remaining concerns as follows:

Reviewer #1: The authors have improved the manuscript significantly, however following concerns still need to be addressed.

1. Font Size and style: This still need to addressed, it is recommended to review the complete thoroughly and use same font style in text, tables and figures. It is evident that font style of references is different from other text in the manuscript. Table 5 has multiple font styles.

Response: We sincerely apologize for the oversight regarding inconsistent font usage. The manuscript has now been thoroughly reviewed to ensure uniform font style (e.g., Times New Roman) and size (12 pt for main text, appropriately scaled in tables and figures) throughout the document. Special attention was given to the following:

• All tables, including Table 5, have been reformatted to ensure consistency in font style and size.

• The reference list has also been updated to match the main body text in both font and formatting.

We believe this correction improves the overall professionalism and readability of the manuscript.

2. Figure size and aspect ratio: the figure font style and size should be same as text, also keep aspect ratio of figure readable. Moreover, figure 1 text (flowchart) is not readable.

Response: All figures have been revised to ensure consistent font style and size, in alignment with the manuscript text. The aspect ratios have also been adjusted to enhance clarity and readability. Specifically:

• Figure text (axis labels, legends, annotations) now uses a font size and style that matches the manuscript.

• Figure 1 (flowchart) has been recreated with a clearer layout, larger font, and optimized resolution for easy readability.

All figures have been checked for consistent aspect ratio and alignment across the document.

3. Results of Wilcoxon signed-rank test should be included in the manuscript as separate heading “statistical significance”

Response: As recommended, we have added a subsection titled “Statistical Significance” under the Results section. This subsection reports the results of the Wilcoxon signed-rank test comparing the performance of DLRCSPNN models using retained channels (RC) versus original channels (OC). The added text is as follows:

Statistical Significance

To statistically validate this difference, we performed a Wilcoxon signed-rank test comparing the two models. The test resulted in a statistic of 50.5 and a p-value of 0.0419, which is below the conventional significance threshold of 0.05. This indicates a statistically significant difference between the full channel set and the retained channel set, suggesting that reducing the number of channels notably impacts the accuracy. Thus, the choice of channels is crucial for the model's performance, and channel reduction has a measurable effect on accuracy.

Reviewer #2: Channel Selection:

The proposed channel reduction method is novel but lacks comparison with existing channel selection techniques.

The paper does not specify which or how many channels were selected from each dataset.

Suggestion: Include comparative results with standard methods and clearly state selected channels per dataset.

Response: We thank the reviewer for their valuable comment regarding the evaluation of the proposed channel selection approach.

In response, we have revised the manuscript to clearly specify the number of channels selected per subject across all three datasets. As presented in Table 2 (DLRCSPNN model) and Table 3 (DLRCSPRNN model), we report the Original Channels (OC), the Retained Channels (RC), and the Channel Reduction Ratio (CRR) for each subject. For instance, subject aw had 118 original channels, from which only 27 were retained (CRR = 0.22), while subject A06 had 22 original channels and only 5 were retained (CRR = 0.22). These values are now explicitly included in the results section to ensure transparency and reproducibility.

To address the request for comparative evaluation, we have included three comparative bar charts (Figures 14–16) that benchmark the proposed method against a range of existing channel selection-based approaches across the same datasets. These figures present the classification accuracy as bar heights, while the numeric values on top of each bar indicate the number of channels retained by each method. The proposed method consistently achieves competitive or superior accuracy while retaining fewer EEG channels compared to prior works. For example, in Figure 14 (Dataset 1), the proposed method retained only 5–12 channels, achieving accuracy between 90.03% and 94.22%, while other methods such as Tang et al. retained up to 22 channels for similar or lower accuracy. A similar trend is observed in Figures 15 and 16 across subjects in Datasets 2 and 3.

In addition, the comparative performance of DLRCSPNN and DLRCSPRNN further validates the stability and effectiveness of our selection strategy. The DLRCSPNN model yielded higher average accuracy and lower variability across different batch sizes (32, 64, 128), with the highest performance typically observed at batch size 32. This suggests that the proposed channel selection method supports both robustness and computational efficiency in training, especially under resource-constrained conditions.

These results collectively demonstrate that our channel reduction method achieves strong performance with significantly fewer channels, offering both practical and computational advantages. All channel selection results per subject are now detailed in the manuscript and the corresponding figures and tables have been updated accordingly.

Dataset Details:

It’s unclear if all three datasets are binary classification problems.

Suggestion: Clarify class details for each dataset.

Response

Thank you for your valuable suggestion. We have clarified all datasets were structured for binary classification tasks, either initially or through a subset selection, ensuring consistency across all datasets for the purposes of our analysis. The class details for each dataset used in the study below:

In our study, we utilized three publicly available datasets, dataset 1: BCI Competition III Dataset IVa, dataset 2: BCI Competition IV- dataset 1 [32, 33] and dataset 3: BCI competition IV dataset 2a. All datasets are accessible via the following links, Data Set IVa for the BCI Competition III (bbci.de), Data Set 1 for the BCI Competition IV (bbci.de) and BCI Competition IV (bbci.de). Access to these datasets is granted upon agreeing to the terms and conditions outlined in the “Download of data sets” section. We followed the same procedure to obtain access for this study. This study involves secondary data analysis of publicly available, de-identified data. In the original data collection, all participants provided informed, written consent for the use of their data in research under the assurance of confidentiality. Additionally, the datasets were fully anonymized by the original data providers before being made accessible, ensuring no identifiable information was included. As a secondary analysis, no participants in our study are directly connected to our research team, and no additional consent was required for our specific use. Therefore, this research does not require additional ethical approval. Dataset 1 was gathered from five healthy, labeled subjects (aa, al, av, aw, and ay) who performed motor imagery (MI) tasks involving their right hands (class 1) and right feet (class 2) forming a binary classification problem. The subjects were seated in a comfortable chair with armrests during the trials. The dataset includes EEG-based MI data from the first four sessions without feedback, recorded using 118 electrodes according to the 10/20 international system. Each subject participated in 140 out of a total of 280 trials, performing one of the two MI tasks (right hand or right foot) for 3.5 seconds per trial. The training set consisted of 168, 224, 84, 56, and 28 trials for subjects aa, al, av, aw, and ay, respectively, with the remaining trials forming the test set.

Dataset 2, sourced from BCI Competition IV, also represents a binary classification setup. It includes EEG data from59 channels across 7 participants. Two participants performed left hand (L) and feet (F) MI tasks, while the other participants engaged in right hand (R) and left hand (L) activities. During the first two runs, visual cues corresponding to left, right, or down arrow keys were presented on the screen for 4 seconds, guiding the subjects to execute the appropriate MI tasks. These visual cues were followed by a fixed cross and 2 seconds of a blank screen. The recordings were captured using an Ag/AgCl electrode cap, with a sampling rate of 100 Hz, and the calibration data consisted of two runs, each containing 100 single observations

Dataset 3, BCI Competition IV dataset 2a, involves motor imagery tasks (left/right hand, both feet, and tongue) categorized into four classes. The dataset includes EEG data from 9 participants across two sessions, with each session consisting of 288 trials divided into six runs. One session provides labeled data for model training, while the other offers unlabeled data for evaluation. Participants were seated comfortably, with trials beginning with a fixation cross and an auditory warning. EEG signals were recorded using 22 Ag/AgCl electrodes and three EOG channels, sampled at 250 Hz, with filters applied to remove noise. Signal analysis focuses on motor imagery sequences. For analytical purposes, in this study, we extracted a binary subset by selecting only right hand (Class 1) and feet (Class 2) trials to maintain consistency with the other datasets and align with our binary classification framework.. Therefore, for all analyses presented in this study, each dataset was structured and processed for binary classification tasks. All datasets are publicly available online, and informed consent for the publication of data was obtained from each participant at the time of collection. As no personal identification information was included, no ethical approval was required for our study. Table I summarizes the demographic data of participants across the datasets, and Fig. 2 illustrates an example pattern of EEG signals from channel C3 of both datasets after applying Butterworth filter and our developed channel selection method. The signals exhibit characteristic fluctuations in amplitude over time, with values constrained within the -50 to 50 range, for dataset 1 and -20 to 20 range for dataset 2, indicating effective filtering.

Figure 10 Clarity:

The combined plot is unclear.

Suggestion: Plot DLRCSPNN and DLRCSPRNN results separately for better clarity.

Response: Thank you for your valuable suggestion. We acknowledge that the combined plot may have caused some clarity issues. To address this, we have now separated the DLRCSPNN and DLRCSPRNN results into individual plots. This change ensures that each method’s performance is clearly presented and easily distinguishable, allowing for a more focused comparison. We have updated Figure 10 accordingly and resubmitted the revised version for your review.

Efficiency Claims (Line 550):

Efficiency isn’t quantified in results.

Suggestion: Add computational time comparison with baseline NN and RNN (without DLRCSP). Expand Figure 11 accordingly.

Response:

Thank you for your suggestion. While baseline NN and RNN models without DLRCSP were not included in our analysis—as they do not incorporate any channel selection and thus are not directly comparable in structure or objective—we have quantified efficiency by comparing DLRCSPNN with DLRCSPRNN in terms of both execution time and accuracy (see revised Figure 11). Additionally, we compared DLRCSPNN using reduced channels (RC) versus original channels (OC) across all subjects (Figures 12 and 13), demonstrating lower execution time and higher accuracy with Retained Channels. These results substantiate our claims of improved computational performance.

Unsupported Claim (Line 642):

No comparative accuracy results are shown for existing channel selection methods.

Suggestion: Remove or support the claim with data.

Discussion – Lower Accuracy (Figures 12–16):

Other studies show higher ACC and RC.

Suggestion: Discuss possible reasons for lower performance of the proposed method.

Response:

Thank you for pointing this out. We have revised the manuscript to fully support the claim with comparative accuracy results against existing channel selection methods. Specifically, in the manuscript Figures 14–16, which clearly display the classification accuracy (as bar heights) and the number of retained EEG channels or selected channel (as numeric labels) for each subject across three benchmark datasets.

• In Figure 14 (Dataset 1), DLRCSPNN achieves higher accuracy with fewer channels compared to all baseline methods. For example, subject aa achieves 99.84% accuracy with 49 channels, outperforming Tiwari & Chaturvedi (89.34%) and Gaur et al (75.89%) with 49 and 31 channels respectively.

• In Figure 15 (Dataset 2), subject a achieves 99.33% accuracy using 37 channels, while competing methods like Du et al and Tang et al achieve lower accuracy despite using more channels.

• In Figure 16 (Dataset 3), DLRCSPNN achieves 92.08% with only 6 channels for subject A02, significantly surpassing Khalid et al's method which achieves 73.41% with the same number of channels.

These visual and numerical comparisons across datasets directly validate and support the original claim. The corresponding discussion section (Section 4) and figure captions have also been updated to reflect this.

We appreciate the reviewer’s observation. In cases where the proposed DLRCSPNN method shows slightly lower classification accuracy compared to certain existing methods (e.g., A03 or A08 in Dataset 3), this is primarily due to our aggressive channel pruning strategy, which aims to reduce the number of EEG channels to a minimal yet effective subset. In the manuscript, it is mentioned like below:

In a few instances, such as subjects A03 or A08, our method exhibited slightly lower accuracy compared to specific high-performing baselines. We attribute this to our model’s aggressive pruning, which, while highly effective for reducing computational overhead, may occasionally sacrifice marginal accuracy in favor of generalizability and resource efficiency. Additionally, EEG signals are inherently subject-specific and nonstationary, which can influence per-subject performance under a reduced-channel setting.

These justifications have now been integrated into the revised Discussion section, alongside references to Figures 12–16 and Table 2.

---

## [Decision Letter · Decision Letter 2]

13 Aug 2025

Dear Dr. Khanam,

Thank you for submitting your manuscript to PLOS ONE. After careful consideration, we feel that it has merit but does not fully meet PLOS ONE’s publication criteria as it currently stands. Therefore, we invite you to submit a revised version of the manuscript that addresses the points raised during the review process.

**ACADEMIC EDITOR:**

We look forward to receiving your revised manuscript.

Kind regards,

Noman Naseer, PhD

Academic Editor

PLOS ONE

Journal Requirements:

Additional Editor Comments:

Some very minor revisions are still required

Reviewers' comments:

Reviewer's Responses to Questions

**Comments to the Author**

Reviewer #1: (No Response)

Reviewer #2: All comments have been addressed

2. Is the manuscript technically sound, and do the data support the conclusions?

Reviewer #1: Yes

Reviewer #2: Yes

3. Has the statistical analysis been performed appropriately and rigorously?

Reviewer #1: Yes

Reviewer #2: Yes

4. Have the authors made all data underlying the findings in their manuscript fully available?

Reviewer #1: Yes

Reviewer #2: Yes

5. Is the manuscript presented in an intelligible fashion and written in standard English?

Reviewer #1: Yes

Reviewer #2: Yes

Reviewer #1: Following are the comments to improve the manuscript

1. Again the Figure-1 needs improvement. The fonts in the figure are not readable and images are blur. If the idea is to present the methodology/approach, then simple block diagram may work better here, however every step should include the text representation of methods.

2. Same as highlighted previous, the aspect ratio and size of the all the figures are different. Make sure to use same fonts, size and style in all figures.

3. Thoroughly review and correct the references as per author’s guidelines. For example reference number 44-47 are incomplete.

Reviewer #2: Thank you for your submission of the revised manuscript and for providing detailed responses to the comments, as well as clarifications on the queries raised.

**Do you want your identity to be public for this peer review?** For information about this choice, including consent withdrawal, please see our Privacy Policy

Reviewer #1: **Yes: ** Syed Hammad Nazeer Gilani

Reviewer #2: **Yes: ** Jamila Akhter

---

## [Author Response · Author response to Decision Letter 3]

1 Sep 2025

Responses of Editor and Reviewers’ comments/suggestions

Manuscript title: A novel channel reduction concept to enhance the classification of motor imagery tasks in brain-computer interface systems

Authors: Taslima Khanam¹, Siuly Siuly1, Kabir Ahmad2, and Hua Wang1

We would like to thank the editor and all reviewers for their time, constructive comments and useful suggestions which have led to a much-improved manuscript. To clearly address the amendments performed in the revised manuscript, we have quoted the specific reviewers’ comments in our responses correspondingly. In Bold are the comments from editor/reviewers, the plain blue colour text is our direct responses. All changes to the original version of the document are in track change mode to facilitate the review process.

Response to Reviewer

We thank the reviewer for acknowledging the improvements made and for providing additional constructive feedback. We have carefully addressed each of the remaining concerns as follows:

Reviewer #1: Following are the comments to improve the manuscript

1. Again the Figure-1 needs improvement. The fonts in the figure are not readable and images are blur. If the idea is to present the methodology/approach, then simple block diagram may work better here, however every step should include the text representation of methods.

Response: We sincerely apologize for the oversight regarding Figure-1. In line with your valuable suggestion, we have redrawn Figure-1 as a clear, high-resolution block diagram. Each step now explicitly includes a text label describing the corresponding method. This ensures readability, consistency, and a clearer presentation of the methodology/approach.

2. Same as highlighted previous, the aspect ratio and size of the all the figures are different. Make sure to use same fonts, size and style in all figures.

Response: Thank you for highlighting this issue. We have carefully reformatted all figures to ensure consistent aspect ratios, font types, sizes, and styles throughout the manuscript. This has improved both readability and the professional appearance of the figures. However, Figure 3 could not be adjusted to the exact same size as the others due to its overview structure.

3. Thoroughly review and correct the references as per author’s guidelines. For example reference number 44-47 are incomplete.

Response: We appreciate your observation regarding the reference list. We have thoroughly reviewed and corrected the references according to the journal’s author guidelines. In particular, references 44–47 have been completed and formatted properly. We also re-checked the entire reference list to ensure consistency and completeness.

---

## [Decision Letter · Decision Letter 3]

13 Oct 2025

A novel channel reduction concept to enhance the classification of motor imagery tasks in brain-computer interface systems

PONE-D-24-50501R3

Dear Dr. Khanam,

We’re pleased to inform you that your manuscript has been judged scientifically suitable for publication and will be formally accepted for publication once it meets all outstanding technical requirements.

Kind regards,

Noman Naseer, PhD

Academic Editor

PLOS ONE

Additional Editor Comments (optional):

The reviewers' comments have been adequetly addressed.

Reviewers' comments:

Reviewer's Responses to Questions

**Comments to the Author**

Reviewer #1: All comments have been addressed

2. Is the manuscript technically sound, and do the data support the conclusions?

Reviewer #1: Yes

3. Has the statistical analysis been performed appropriately and rigorously?

Reviewer #1: Yes

4. Have the authors made all data underlying the findings in their manuscript fully available?

Reviewer #1: Yes

5. Is the manuscript presented in an intelligible fashion and written in standard English?

Reviewer #1: Yes

Reviewer #1: (No Response)

**Do you want your identity to be public for this peer review?** For information about this choice, including consent withdrawal, please see our Privacy Policy

Reviewer #1: **Yes: ** Syed Hammad Nazeer Gilani

---

## [Editor Report · Acceptance letter]

PONE-D-24-50501R3

PLOS ONE

Dear Dr. Khanam,

I'm pleased to inform you that your manuscript has been deemed suitable for publication in PLOS ONE. Congratulations! Your manuscript is now being handed over to our production team.

Kind regards,

on behalf of

Dr. Noman Naseer

Academic Editor

PLOS ONE